# Dung removal increases under higher dung beetle functional diversity regardless of grazing intensification

Dung removal by macrofauna such as dung beetles is an important process for nutrient cycling in pasturelands. Intensification of farming practices generally reduces species and functional diversity of terrestrial invertebrates, which may negatively affect ecosystem services. Here, we investigate the effects of cattle-grazing intensification on dung removal by dung beetles in field experiments replicated in 38 pastures around the world. Within each study site, we measured dung removal in pastures managed with low- and high-intensity regimes to assess between-regime differences in dung beetle diversity and dung removal, whilst also considering climate and regional variations. The impacts of intensification were heterogeneous, either diminishing or increasing dung beetle species richness, functional diversity, and dung removal rates. The effects of beetle diversity on dung removal were more variable across sites than within sites. Dung removal increased with species richness across sites, while functional diversity consistently enhanced dung removal within sites, independently of cattle grazing intensity or climate. Our findings indicate that, despite intensified cattle stocking rates, ecosystem services related to decomposition and nutrient cycling can be maintained when a functionally diverse dung beetle community inhabits the human-modified landscape.

Land-use intensification is a major threat to biodiversity, constituting one of the most critical global change effects of the Anthropocene[1,2]. Such landscape changes have driven significant biodiversity losses[3,4]. However, intensified agricultural practices can affect biodiversity and ecosystem services in different ways[5–7]. In some cases, agricultural intensification leads to massive losses in biodiversity[8] with consequent reductions in ecosystem functioning and provision of ecosystem services[9]. In others, the increase in landscape heterogeneity and productivity associated with mosaic farming practices does not cause severe biodiversity losses, and may even increase ecosystem functioning[10]. These heterogeneous outcomes may result from regional differences in agricultural practices and/or the ecological characteristics of the species present in the regional pool. The biotas of different biogeographical regions have been shaped by divergent evolutionary histories and environmental conditions, including climate. Such divergences have led to differences in species pools,

biodiversity patterns, and dynamics of local communities[11,12]. Of particular importance are regional differences in functional diversity, i.e. the components of biodiversity that influence ecosystem functioning, which are typically measured by the variation in the traits that are related to these functions[13]. Regional variations in the ecological characteristics and functional traits of the species pools can impact the relationship between land-use intensification and biodiversity[14]. In consequence, regional differences in biotas, farming intensity and management practices may result in different outcomes for the same ecosystem service in different parts of the world.

Available evidence suggests that three different biodiversity–ecosystem function (BEF) relationships may occur under increasing agricultural intensification. First, intensification may reduce biodiversity and consequently decrease ecosystem functioning[8]. Second, intensification may enhance productivity leading to larger populations of some species, which in turn enhance ecosystem

✉ e-mail: jhortal@mncn.csic.es; anamc.santos@uam.es

function, but do not necessarily increase biodiversity[15]. Third, the level of intensification may be variable across habitat patches and/or time, which may increase overall biodiversity and ecosystem functioning at the landscape level[10]. Determining when and how each scenario may arise is key to improving land management practices through 'ecological intensification' regimes−that is, minimizing environmental impacts while sustaining ecosystem services and agricultural production[5,16]. However, the factors associated with variability in BEF relationships under increasing intensification have been poorly studied, with a general lack of experimental evidence on ecosystem service provision (but see ref. 7), particularly in insects[17]. It is therefore necessary to develop standardized assessments for evaluating the global effects of apparently similar transformations affecting the same type of ecosystems, while simultaneously considering regional differences in climate, species pools and agricultural practices. This is critical for disentangling the factors that determine different impacts of intensification on biodiversity and ecosystem functioning, and identifying similar mechanisms across different biogeographical regions.

We assessed the relationship between grazing intensity, climate, biodiversity, and ecosystem functioning on global pasturelands, a human-transformed ecosystem with many similarities worldwide. Specifically, we studied the determinants of several aspects of dung beetle diversity (abundance, species richness and functional diversity), as well as their effects on the provision of dung removal, a key ecological function associated with several ecosystem services, including nutrient cycling, bioturbation, and secondary seed dispersal[18]. To do this, we performed paired field experiments in pasturelands of 38 localities scattered worldwide, with better coverage of the Americas and Europe, but including sites in Africa, Asia, and Australia (Fig. S1). These experiments compared dung beetle diversity and dung removal rates between pastures subject to low- versus high-intensity management regimes. The high stocking rates characteristic of high-intensity regimes produce changes in soils such as soil compaction, nitrification, and decreased water infiltration[19], which affect the pastures, and may cause other impacts associated with practices such as additional hay supplies, more frequent antibiotic and antiparasitic treatments, the introduction of exotic grasses, irrigation, and use of fertilizers[20]. In each locality, we compared one pasture subject to intensive herding with high cattle densities (> 4 animals/ha) and frequent food provision and antiparasitic treatments, in opposition to another pasture under extensive grazing with low cattle density (<2 animals/ha) and only sporadic supplies of food and veterinary treatments.

Dung beetles are a globally distributed group of decomposers that influence many ecosystem processes through the elimination and burial of manure, thus providing various services such as the decomposition of organic matter, nutrient cycling, control of fly and nematode pest populations that affect livestock, soil aeration and bioturbation, greenhouse gas reduction and secondary seed dispersal[18,21–23]. The role played by dung beetles in the ecosystems has been traditionally studied by classifying them into functional guilds based on their nesting and feeding behavior, which have many effects on how they process dung and interact with the soil[23]. However, recent studies indicate that variations in their functional traits may also drive these processes[18,24]. As variations in morphological traits and behavioral guilds are partly independent of each other, they may have different functional effects and determine ecosystem functions in different ways[18]. Therefore, here we have characterized two different aspects of dung beetle functional diversity, based on (i) functional guilds primarily defined according to behavior, and (ii) morphological traits of known functional effect.

We analyzed results from the field experiments using two different approaches. First, we used piecewise Structural Equation Modeling[25] (piecewise SEM; see Methods) to characterize the set of factors that determine BEF relationships across sites. These factors include different aspects of dung beetle diversity as well as variation due to cattle management intensity and climate. Specifically, we

evaluated whether climate and intensified management practices (i.e., intensified land use including increased stocking rates and the use of anthelmintics to deworm livestock) affect taxonomic and functional diversity, resulting in changes that ultimately impact dung removal. Biogeographical variation was considered by including the biogeographical region as a random factor (see Methods). We then applied a meta-analytical approach to assess the factors underlying the effects of adopting either low- or high-intensity management practices within each site. Previous work show that farming intensification reduces dung beetle abundance and diversity, which in turn diminishes the functional impact of these communities on the ecosystems[26]. This leads to the expectation of lower levels of functioning under intensified cattle farming. However, in some cases high-intensity farming does not show adverse effects on dung beetle communities, in particular under adequate landscape management[27]. Thus, some heterogeneity in the responses of dung beetles to intensive practices might be expected, as is indeed shown by our results. However, although our analyses accounting for biogeographical regions show that dung removal increases with species richness and is also affected by climate differences between sites, the field experiments identify a striking and remarkably consistent positive relationship between functional diversity and dung removal within sites, which is independent of cattle grazing intensity and climate.

## Results

Species richness was significantly higher in low-intensity pastures, but abundance, dung removal and functional diversity showed no overall differences between low- and high-intensity pastures (Fig. 1). Piecewise SEM results indicate that dung beetle species richness and climate have the greatest impacts on dung removal rates across sites ($\beta = 0.45$, $p < 0.01$ and $\beta = -0.27$, $p < 0.05$, respectively; Fig. 2). Strikingly, after controlling for the effects of climate, species richness was not significantly affected by the type of management (either low- or high-intensity), or by the frequency of anthelmintic treatments (Fig. 2; Table S1). Most of the variation in community attributes was explained by biogeographical region (see the large differences between conditional and marginal $r^2$ in Table S1), although species richness has a positive effect on behavioral diversity ($\beta = 0.49$, $p < 0.001$), and climate affects morphological diversity ($\beta = -0.40$, $p < 0.05$). These results were robust to other model specifications, as we found that species richness was still the main predictor of dung removal when we replaced climatic conditions by latitude (Table S2).

We applied a prospective meta-analysis[28] to test the effects of climatic conditions, differences in cattle density, and differences in dung beetle community attributes (including differences in abundance, richness, and functional diversity) on the difference in dung removal rates between management regimes (low-high intensities pastures; see Methods). We found a non-significant weighted mean effect size (Hedges' $g \pm 95\%$ CI $= 0.011 \pm 0.54$, $t = 0.042$; $P = 0.967$) indicating that, overall, dung removal rates were similar between management regimes within sites (see also Fig. 1c). This is contrary to the expected reduction in dung removal rates under high-intensity management. However, we found significant heterogeneity in effect sizes ($T^2 = 2.252$; $Q = 300.48$; df $= 37$; $P < 0.01$; Fig. 3a, b; Fig. S3), and approximately 90% of this heterogeneity could be attributed to differences between the 38 experimental sites ($I^2 = 0.898$; see Methods). According to a meta-regression model, within-site differences in dung removal rates between management regimes were mainly due to differences in dung beetle functional diversity, whereas the effects of climate conditions and difference in cattle density were not significant (Table 1). This result indicates that the difference in dung removal rates between low- and high-intensity pastures tended to be higher with higher levels of functional diversity (Fig. 3c). Note also that $PCA1_{Diversity}$, a variable mainly accounting for functional diversity, can also be interpreted as an effect size, so negative and positive scores on this axis indicate that diversity is greater in high-

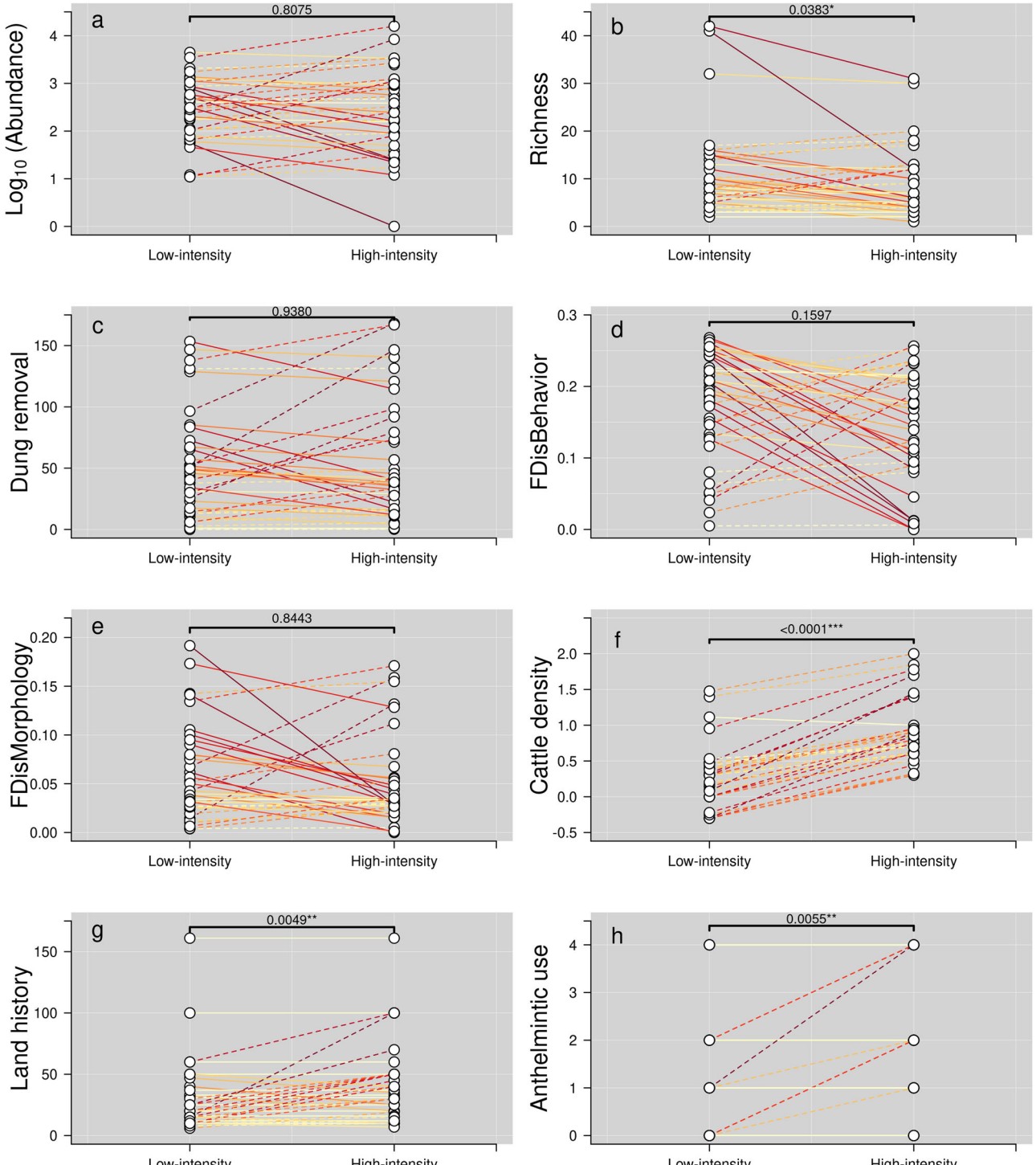

**Fig. 1 | Comparisons of dung beetle diversity and local characteristics within sites between the two levels of farming intensification.** Comparisons between low- and high-intensity management regimes within each site for the whole dataset, including abundance a, species richness b, dung removal rate (g/48 h) c, functional diversity d, e, cattle density (log number of cows/ha) f, land history g, and anthelmintic use h. Each circle represents one of the two field sites (low- vs. high-intensity) within of our paired experimental design. Line colors indicate the magnitude of differences between low- and high-intensity treatments: light yellow indicates weaker differences, while dark red does so for stronger differences.

Continuous lines indicate higher values in low- than in high-intensity treatments, and dashed lines indicate higher values in high- than in low-intensity treatments. *FDisBehavior* and *FDisMorphology* stand for Functional Diversity measured from behavioral and morphological traits, respectively (see Methods). Land history depicts the approximate number of years that the field has been devoted to cattle farming, and anthelmintic use refers to the number of times per year that the cattle are subject to deworming treatments (with, e.g., ivermectins). Values correspond to the significance of a two-tailed Wilcoxon paired test; significant comparisons are highlighted as * = $p < 0.05$, ** = $p < 0.01$, *** = $p < 0.001$. See also Table S1.

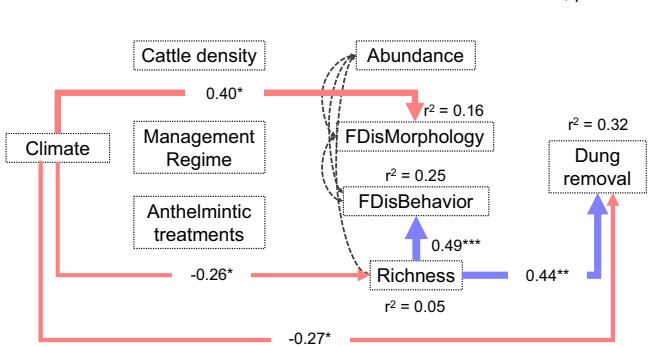

Fisher's C = 20.28; p = 0.44

**Fig. 2 | Results of the piecewise Structural Equation Model showing the putative causal relationships between the factors affecting dung removal rates across sites under different cattle management regimes (low-intensity and high-intensity), taking biogeographical variation into account as a random effect.** Positive and negative effects are indicated by blue and red arrows, respectively, and standardized coefficients (standardized $\beta$) are provided within the arrows; dashed lines indicate non-significant relationships; two-headed dashed black arrows indicate that variables have correlated errors (see Methods); $r^2$ values indicate the level of fit for each fixed factor (i.e., marginal $r^2$); significance values: * = $p < 0.05$, ** = $p < 0.01$, *** = $p < 0.001$; Fisher's C statistic is a measure of how well the model fits the observed data; similarly, a model-wide $P$-value > 0.05 indicates that the observed data supports the hypothesized structure of the model (see Table S1). *FDisMorphology* and *FDisBehavior* stand for the Functional Dispersion of morphological and behavioral dung beetle traits, respectively. See Methods for the origin of all variables.

and low-intensity regimes, respectively. Therefore, the meta-regression results (Fig. 3c) indicate that the effect size of management regime is mainly controlled by functional diversity. If functional diversity is higher in the high-intensity pasture, then the removal rate is higher under this regime (negative g values). Similarly, if functional diversity is higher in the low-intensity pasture, then the removal rate is higher under this regime (positive g values). Consequently, the higher the functional diversity, the higher the dung removal rate independently of the management regime. Our results were robust to other model specifications, as we found that the difference in functional diversity was still the main moderator when the meta-regression models were estimated using different attributes of dung beetle communities separately (Table S4) or when we replaced climate conditions by latitude (Table S5). Although these two alternative models were similar in terms of explanatory capacity, the climatic model was more robust, so given that its effects on dung beetle diversity are extensively supported by the literature, we used climate over latitude as a moderator of diversity. Note also that the apparent contradiction between meta-regression and SEM results is due to the basic differences between both analyses. While SEMs assess variations between sites, treating each one of the sites of the paired experiment in a landscape as a sampling unit, the meta-analysis assessed differences in dung removal rates within sites, i.e. between types of management within each location.

## Discussion

The results of our experiments on dung removal by dung beetles demonstrate that greater species richness and functional diversity enhance ecosystem functioning, but their effects may occur at different scales. Importantly, there was a large between-region heterogeneity in the biodiversity–ecosystem functioning (BEF) relationship. However, once such heterogeneity is accounted for, within-site differences in dung removal between low- and high-intensity management regimes are driven by dung beetle functional diversity. This shifts the focus of BEF maintenance beyond the local communities, towards the landscape and regional scales, providing a mechanistic explanation

for the recent finding that grazing intensification can be detrimental in species-poor and warm arid zones, and beneficial in milder and species-rich areas[7]. Thus, our findings provide a deeper understanding of these heterogeneous effects of increasing intensification of cattle farming in different biogeographical regions. Because the functional structure of communities is largely determined by their species pool[29], our results imply that community-level responses to intensification are determined by the functional characteristics of the species present in each landscape[30], and may be influenced by the connectivity of habitat patches within these landscapes[31]. Considering that human actions have transformed most terrestrial ecosystems, the current distribution of species and functional diversity has been driven by this history of land occupation and anthropogenic transformation[32,33]. Following a response–effect trait framework[34], the differences in the traits of the species present in each landscape may trigger different responses to these human-induced historical transformations, thereby leading to heterogeneous effects of intensification on ecological functioning.

Beyond the importance of functional diversity for enhancing dung removal within sites, the SEM analyses indicate that species richness and climate are the main regulators of this ecological function across sites throughout the world. This implies that greater species richness can maintain ecosystem service provision by dung beetles even under intensified treatments, a pattern also found in other perturbed agro-ecosystems[35]. However, the relationship between climatic gradients and dung beetle diversity can determine the outcome of the ecological functions performed by these insects[36], so maintaining multiple species may buffer the effects of climate change in perturbed systems[37]. Although fragmented landscapes may be able to maintain high levels of dung beetle species richness, these species often show reduced trait variation[38]. So, when the pool of dung beetle species adapted to exploit the feces of ungulates is naturally poor, or has been impoverished by human-induced extinctions, the advantages of maintaining more diverse communities locally through low-intensity management practices may be smaller, due to the limited functional diversity available in a poor species pool. Clearance of native vegetation and intensification of land use are known to reduce both dung beetle species richness and their role in ecosystem functioning[39,40]. However, in our study, local factors related to intensification such as the application of anthelmintic treatments or the history of land transformations had no significant effect on dung removal at the global scale of our SEM analyses. If local effects were the main drivers of ecosystem functioning, they would show consistent effects worldwide, favouring higher dung removal rates in either low- or high-intensity grazed pastures according to the conditions present in each location. Yet, these local factors seem to have little global effect across sites. Other factors that could potentially affect dung removal rates, such as weather conditions during fieldwork, were controlled for by the experimental protocol (see Methods), so their effect should be minimal and cause only small stochastic variations in the data. In consequence, the heterogeneity in the response to intensification may be due to both differences in the regional species pool, and the widely different practices that are applied in intensified cattle management in different regions or countries.

The effects of cattle density and dung beetle abundance were not significant in the SEM analyses. Although it is possible that in some cases, dung removal in high-intensity regimes is maintained by a few hyper-abundant species, this is not supported by our data. Here it is important to note that the effects of dung beetle abundance on dung removal are largely species-specific[18,41]. Nonetheless, our study shows positive effects of intensification on dung removal in regions where dung beetle functions are driven by a few species, such as Australia (Fig. 3a), which lacks a native dung beetle fauna well-adapted to cattle dung[42]. So, in the novel Australian pasture ecosystems, dung removal relies on a small suite of recently introduced species[43]. A small group of species is also responsible for dung removal in the UK and Central and

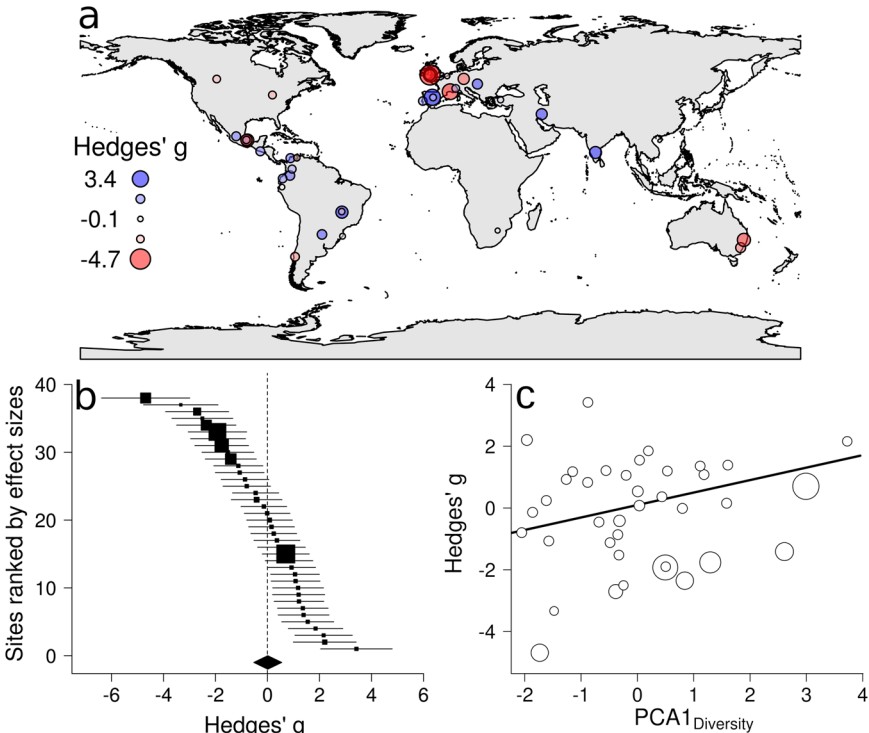

**Fig. 3 | Effect sizes of grazing intensification on the ecosystem services provided by dung beetles in 38 landscapes worldwide.** Values of standardized mean differences (Hedges' *g*) indicate the within-site differences in dung removal between pastures subject to low- or high-intensity cattle management regimes in the same site. Negative Hedges' *g* values indicate greater dung removal in high-intensity pastures, and positive values indicate greater removal in low-intensity pastures. **a** Geographical distribution of the studied locations; circle sizes indicate absolute Hedges' *g* values; blue circles indicate higher dung removal in low-intensity and red circles in high-intensity study sites, respectively. **b** Forest plot showing the effects of cattle management regime (low- vs. high-intensity) on dung removal rates in the 38 studied landscapes. Squares indicate the estimated effect sizes (Hedges' *g*) at each landscape and bars indicate their 95% confidence intervals estimated following a random-effects model. Sizes of the squares indicate the weights of each effect size for the weighted mean effect size estimate (closed

diamond). The width of the diamond represents the 95% confidence interval of the weighted mean effect size. The vertical dashed line indicates an effect size equal to zero. **c** Relationship between dung beetle functional diversity (*PCA1_Diversity*) and within-site differences in dung removal rates between low- and high-intensity cattle management systems (as given by Hedges' *g*). The continuous line indicates the predicted values of the random-effects meta-regression model. Circle sizes are proportional to the weights of the effect sizes. *PCA1_Diversity* was based on differences in diversity measures between low- and high-intensity systems (Functional Dispersion based on behavioral and morphological traits; *FDisBehavior* and *FDisMorphology*). Thus, as the differences calculated using the diversity indices were positively correlated with the first PCA axis, positive scores indicate higher diversity in low-intensity systems, and negative values indicate higher diversity in high-intensity systems.

Eastern Europe, where Holocene land-use changes have resulted in an acceleration of dung beetle extinctions[44], a trend exacerbated by the wide adoption of insecticide and anti-parasitic treatments[45]. In these two cases, functioning is mainly driven by abundance[41], implying that

the increasing manure production of more productive farms is processed by high numbers of a small set of dung beetle species.

Although the provision of ecosystem services by dung beetles is known to depend on both species richness and functional diversity[46,47], the results of our meta-analyses show that increasing levels of dung beetle functional diversity provide higher levels of dung removal that go well beyond any gains driven by higher species richness (see Table S3). According to the classical view of BEF relationships (i.e. higher biodiversity enhances ecosystem functioning), the adverse effects of cattle herding intensification on dung beetle diversity may produce a cascading effect that leads to a general reduction in functioning associated with high-intensity cattle management as found by our prospective meta-analysis. But our results also show that the negative effect of intensification on dung removal rates can be exacerbated if there is a selective loss of species with particular sets of traits, as indicated by the importance of functional diversity metrics in this same analysis. This provides evidence to support that maintaining a functionally rich species pool within a diverse landscape is key to sustaining higher levels of functioning, and thus higher ecosystem service provision. Indeed, dung beetle functional diversity and ecosystem services are enhanced by management strategies of landscapes that maintain remnants of native vegetation and apply low levels of anti-parasitic cattle treatments[27,35]. For different reasons, this is the

**Table 1 | Results of a meta-regression model assessing the effects of climate (*PCA1_Climate*) and variation in cattle density (*Δ Cattle Density*) on the difference in dung removal rates between pasture management regimes (low-high intensities) within sites**

| Moderators | Estimate | SE | t | df | P | VIF |
|---|---|---|---|---|---|---|
| Intercept | 0.097 | 0.261 | 0.372 | 34 | 0.712 | |
| *PCA1_Climate* | −0.098 | 0.103 | −0.959 | 34 | 0.344 | 1.05 |
| *Δ Cattle Density* | −0.185 | 0.239 | −0.775 | 34 | 0.444 | 1.01 |
| *PCA1_Diversity* | 0.404 | 0.161 | 2.502 | 34 | 0.017 | 1.07 |

The effects of the other variables (differences in dung beetle abundance, richness, and functional diversity based on both behavioral and morphological traits) were tested altogether as the scores of a principal component analysis that summarizes them (*PCA1_Diversity*; see Table S5). Pseudo-$R^2$ = 0.137; $F_{3,34}$ = 3.30; $P$ = 0.032. VIF stands for variance inflation factor. See also Tables S3 and S4 for comparison with other model specifications. *t*: *t*-test values for the intercept and each partial regression coefficient.

case in several landscapes of Southern Europe or America (Fig. 3). In the Mediterranean basin, the long history of human occupation and clearance of woody plant communities, has promoted a higher diversity of dung beetle species associated with open grasslands[48], which play a major role in the functioning of low- intensity grazing management regimes. In America, dry semi-open savanna-like biomes such as the *Cerrado*, *Chaco* and *Matorral* also hold a diverse assemblage of dung beetles adapted to exploit the feces of the native fauna in more open areas. Due to this fact, although the dung beetles inhabiting pastures show overall lower diversity in these novel open habitats[49,50], they are often functionally diverse when compared to the species able to exploit open habitats in hyper-diverse rainforest biomes such as the Atlantic forest[51]. Such higher level of native functional diversity adapted to open habitats in savanna-like biomes[49] may be able to sustain service provision in low-intensity management pastures more efficiently than the functionally poorer pool of species adapted to open habitats inhabiting forest biomes. Nonetheless, in regions that were originally largely forested, the communities in the novel open habitats will be composed of ecologically similar species, following a phylogenetically and functionally structured pattern of biotic homogenization[52]. Therefore, in originally forested biomes, dung beetles may show higher dung removal rates in the native forests than in the open forest gaps and novel pastures, both in northern Europe[53–55] and American tropical forest biomes[40,56,57].

Biodiversity–ecosystem functioning relationships are generally considered to be positive[58]. However, they are variable in natural systems, and can become negative depending on the differences between the species pool and local diversity[59]. There is a consensus that functioning is enhanced by high complementarity between species and functional diversity[60], because communities with higher diversity are more likely to include species that perform different functions[61]. However, there are some difficulties in disentangling the effects of species richness versus functional diversity[62]. Although our results show that functional diversity plays a key role in BEF relationships, the high abundance of some, arguably, functionally redundant dominant species may be also important when cattle densities are high. This calls for adopting a pragmatic view on land use intensification in heavily transformed landscapes. Rather than assuming that intensive practices will inevitably have adverse effects on ecosystem functioning, they may enhance functioning in landscapes inhabited by functionally-poor species pools by expanding their overall abundance. Thus, while some aspects of intensification can be detrimental for the delivery of some ecosystem services in one area, they may bear no effects or even enhance the same services in others, depending on the functional composition of their species pools. This implies that the effects of different management practices on biodiversity and BEF relationships should be evaluated in the context of the pool of resident species performing each essential function in every region or landscape.

Increasing ecosystem functioning may require strategies for increasing functional diversity at the landscape level. Such an approach would demand the design of effective local adaptive management under an 'ecological intensification' framework that sustains ecosystem services and food production while minimizing impacts[16]. Landscape heterogeneity is already known to improve the potential benefits of organic farming for biodiversity and ecosystem service provision by other species groups[63]. These strategies may eventually include changes in landscape structure, as well as the (re)introduction of species to promote higher levels of functional diversity. Although this strategy has been mostly focused on large mammals, it could be relatively easily and affordably extended to many key invertebrate species that act as ecosystem engineers. For example, introduction programs have been successfully used to establish populations of exotic dung beetles in new regions, such as Australia, to accelerate dung degradation where the native dung beetle fauna is not adapted to use livestock dung[64]. It remains to be seen if a similar relocation approach can be successful

for recovering populations of native dung beetle species in their original habitats (see https://rewildingeurope.com/news/dung-458beetle-release-highlights-the-key-role-of-small-critters-in-rewilding/, for a recent attempt). For these strategies to successfully enhance the biodiversity of intensively managed landscapes over the long term, they need to be part of novel management scenarios where devoting a significant number of land patches to low-intensity practices will provide progressively larger benefits thanks to their contribution to a higher diversity of the landscape species pool, thus reaching a new balance between ecosystem functioning and productive human activities.

## Methods
### Dung removal experiments
Dung removal rates were measured in 2016 and 2017 through field experiments in 38 pairs of sites during the highest peak of yearly dung beetle activity, which varies between regions (see Supplementary Data S1). Study sites correspond to pastures subject to low- and high-intensity management regimes, respectively characterized by densities <2 cows/ha or > 4 cows/ha. These paired field experiments were conducted in open pastureland landscapes where both types of management were present, selecting pastures located within 15 km from each other. However, the levels of stocking used for intensive and extensive herding vary across the globe, according to climate and regional practices (see ref. 24). Therefore, we chose a priori these two cattle density thresholds based on the stocking rates most commonly used by farmers. We combined both types of management to distinguish unequivocally between low- and high-intensity ranching in dry pastures of Mediterranean and subtropical regions, assuming that regions with higher levels of precipitation would allow much higher stocking levels in high-intensity pastures.

Dung removal rate was quantified in each study site through standardized field experiments[65] consisting of a 450 m-long linear transect with ten experimental units separated 50 m from each other (following ref. 66). Each experimental unit consisted of 300 g of fresh cattle dung, placed directly on the soil surface. Five control units were also set up to evaluate dung weight loss by evaporation, consisting of 300 g of fresh cattle dung, placed 50 m apart from the experiment transect, directly on the ground, and covered with a fine polyester mesh fabric (mesh size <2 mm) to prevent access to dung beetles. Fecal excretion of certain veterinary parasiticides can affect dung beetle activity[67]. For this reason, care was taken to use only dung from cattle that had not been treated with these parasiticides in the previous two months. Each experimental and control dung pat was weighed before the experiment. These experimental units were left in the field for 48 h. After the experiment, the dung pats were first cleared of any attached sand, earth, pebbles or pieces of vegetation. Then, they were weighed to estimate their final wet weight, and finally stored in paper bags. These samples were later dried in an oven at 80 °C for 72 h to obtain their dry weight[68]. Dung removal rate (*DRR*) was calculated for each experimental unit using the following set of equations:

$$WP = (FWwet - FWdry)/FWwet \qquad (1)$$

$$IWdry = IWwet*(1 - WP) \qquad (2)$$

$$DRR = (IWdry - FWdry) \qquad (3)$$

where *WP* is water proportion (evaporation in the field), *FW* is the final experimental dung weight (either wet or dry), and *IW* is the initial experimental dung weight (wet or dry).

### Dung beetle surveys
Dung beetle communities were surveyed immediately after removing the experimental units to ensure that climatic conditions−and thus

dung beetle activity—were similar. At each site, we placed ten baited pitfall traps along a 450 m linear transect that were separated from each other by 50 m[66]. Each trap consisted of a 3 L plastic bucket buried at ground level and filled with 1.5 L of preservative fluid (water + soap + salt). It was covered with a 25 × 25 cm plastic or metallic chicken net (2 × 2 cm mesh size) pinned down to the ground[65]. A 300 g bait of fresh, homogenized, and anthelmintic-free cattle dung, was placed on top of this mesh. In some localities, a white plastic dish covered the trap to prevent dung disintegration due to intense rainfall. Samples were collected after 48 hours. At each site, we measured cattle density (i.e., number of cattle per hectare) and obtained information on the frequency of anthelmintic use (i.e., number of applications per year, ranging from 0 to a maximum of 4), land-use history (i.e., number of years of continuous use of the site for cattle grazing), and other local characteristics, through interviews with the farmers (Supplementary Data S1). We also characterized the climate of each pair of sites through 13 bioclimatic variables of Worldclim 2.0[69]. More details on the field and laboratory protocols can be found in ref. [65].

Dung beetles comprising Geotrupidae and Scarabaeidae (including subfamilies Aphodiinae and Scarabaeinae) were identified to the level of species or morphospecies using regional monographs and taxonomic keys. Each species was assigned to one of the traditional four guilds defined by food relocation behavior[70]: paracoprids (tunnelers), telecoprids (rollers), endocoprids (dwellers), and kleptocoprids (kleptoparasites). Species classification into these groups was based on the expert knowledge of the dung beetle specialists in each field group (all of them authors of this paper), as well as in specialized literature from each study area. Each of these four guilds was combined with the average body size of each species (large: >18.0 mm; medium: 18.0 mm to 10.0 mm; and small: <10.0 mm) to create a combination of 12 different functional groups (i.e., large, medium, and small paracoprids, telecoprids, endocoprids, and kleptocoprids; as in ref. [71]).

## Functional diversity measurements

A number of dung beetle traits have been shown to relate to dung removal (see ref. [23] for a review). The limited number of studies and the nature of our experimental design prevent an assessment of whether the effects of single traits on this ecosystem function vary between management regimes. However, we can assess their joint effects through composite metrics of functional diversity. Functional diversity was measured at each locality from ten randomly chosen individuals of each species using nine morphological traits that are potentially associated with dung removal: (i) head length, (ii) head width, (iii) pronotum length, (iv) pronotum width, (v) pronotum height, (vi) elytra length, (vii) protibia length, (viii) protibia width, and (ix) metatibia length[64]. When fewer than ten individuals of a species were collected, all were measured. Morphological measures were performed with a digital caliper (± 0.01 mm). Finally, all individuals measured were dried at 80 °C for 72 h to measure their dry biomass with a precision scale (± 0.01 g). See Fig. S3 and Supplementary Data S5 for an overview of trait values, and Supplementary Data S6 for the values of all individuals measured.

For each site, assemblage diversity was measured through species richness, abundance, evenness, and two functional diversity indices. Specifically, Functional Dispersion (FDis) was estimated as the mean pairwise distance between all species present at a given site (adapted from mean phylogenetic diversity, a metric used initially in community phylogenetic studies[72]). FDis was used to calculate two different indices, based on two different sets of traits: (i) combining the nine morphological traits and species' biomass—*FDisMorphology*; and (ii) combining the functional groups with the individuals' body length (measured as head length + pronotum length + elytra length)—*FDisBehavior*. In the case of *FDisBehavior*, the phenetic tree for functional groups was tailored to incorporate behavior at the root of the tree, with a first split between kleptocoprid/endocoprid and paracoprid/

telecoprid behaviors, which are known to be related. A second split in each branch generated a tree with four branches corresponding to the four main behaviors, followed by the size classes at the tips. We have used these two approaches because some morphological traits and some specific behaviors may be independent of each other, eventually producing different functional effects[18]. Note, however, that body size is an integral part of these two metrics, as is biomass in *FDisMorphology*, and body length in *FDisBehavior*. This is because body size is known to be one of the main drivers of dung beetle functionality, being also linked to both variations in morphological traits and feeding behavior[18,23]. In any case, this approach allows exploring functional complementarity between morphology and behavior.

## Structural equation models

A Piecewise Structural Equation Model (piecewise SEM[25]) was used to disentangle the relative importance of climate (summarized by $PCA1_{Climate}$) and management practices (i.e., low- and high-intensity management regimes, as well as detailed data on cattle density and anthelmintic use) on both dung beetle diversity (i.e., abundance, richness, *FDisMorphology*, and *FDisBehavior*), and dung removal rates across sites. Biogeographical variation was considered by using mixed models and including the biogeographical region as a random factor (Table S1). Here, site was not included as a random factor because it would only include two non-independent observations per cluster, due to our paired experimental design. We performed a piecewise SEM model including each treatment level (i.e. type of management regime) as a binary variable (low-intensity = 0 and high-intensity = 1) to investigate the expected change in the response variables (i.e., abundance, richness, *FDisMorphology*, *FDisBehavior*, and dung removal) as land use intensifies. We chose a SEM framework because it allows testing a priori hypotheses about the direct and indirect effects of predictor variables[73]. For instance, climate may simultaneously influence dung removal rates directly (through its effect on dung desiccation or physical destruction by strong rainfall), or through its indirect effects on the four dung beetle diversity metrics, which in turn affect dung removal rates. First, we defined the conceptual model as a set of regressions, representing the relationships between the variables (Fig. S4). Land History was not included in this conceptual model because it was not significant in preliminary trials. In a second step, we applied a *d*-separation test to evaluate the independence of non-linked paths and therefore, the adequacy of the conceptual model against the observational data[74,75]. Third, we added new paths (i.e., statistically significant relationships based on the *d*-separation test) to the initial model until it fitted the observed data (reached by a *d*-separation *p* value > 0.05[75]). New paths were added only when they were biologically plausible[76]. When *d*-separation detected links between variables that could not be explained by a clear causal relationship but were produced by the same underlying process, such as in the case of behavioral and morphological diversity, we set this association as correlated errors (as implemented in piecewiseSEM 2.3.0[77]), acknowledging the correlation without imputing a causal meaning to it[25,76]. Latitude is a known correlate of the geographical gradients of both diversity and climate, so we evaluated the robustness of the SEM models by repeating the analyses using latitude instead of climate, following the conceptual model in Fig. S5. For a full list of terms and a full view of model structures, see Appendices S1 and S2 in the Supplementary Information. Piecewise Structural Equation Models were performed in R environment[78] using the piecewiseSEM package[77].

## Meta-analyses of experimental results

The effect of grazing intensification on dung removal rates within sites was assessed using a meta-analysis framework. The effect size per site was calculated using Hedges' *g*[79]. To estimate *g*, we calculated the differences in dung removal rates between low and high-intensity grazing managements; thus, positive values of *g* indicate that manure

removal rates were higher under low-intensity than under high-intensity grazing management, whereas negative values indicate the opposite. We estimated the weighted mean effect size assuming a random-effects model[78]. We estimated the variance of the effect sizes ($T^2$) and the proportion of this variance that can potentially be explained by moderators ($I^2$) following refs. [79–82].

We used a meta-regression model[79] to evaluate the effects of climate and both the differences in cattle density and diversity measures (between management regimes) on the effect sizes. To avoid collinearity, we summarized the climate and diversity descriptors using two Principal Component Analyses (PCA, one for climatic variables and another for diversity indices). We use only the first axis of each PCA ($PCA_{Climate}$ and $PCA_{Diversity}$) as moderators in our meta-regression model. The first axis of the climatic data ($PCA1_{Climate}$) represented approximately 51.68% of the variability. $PCA1_{Climate}$ was negatively correlated with the lowest temperature of the coldest month, the average annual temperature, the isothermal temperature, the mean temperature of the wetter quarter, and the mean temperature of the driest quarter. Seasonality and the annual range in temperature were positively correlated with $PCA1_{Climate}$ (Table S5). Dung beetle diversity indices were estimated separately for low- and high-intensity cattle grazing. Thus, we first computed the differences between low- and high-intensity regimes for each diversity index. Then, we used a PCA to summarize these data ($PCA_{Diversity}$). The first axis of the diversity data ($PCA1_{Diversity}$) represented approximately 46.78% of the variance in biodiversity differences. $PCA1_{Diversity}$ was positively correlated mainly with the differences in functional diversity indices measured with behavioral traits (FDisBehavior) and morphological traits (FDisMorphology; Table S6). To avoid collinearity, we did not include variation in anthelmintic use as a moderator in the meta-regression model because it was correlated with cattle density, which is already included as a moderator in our meta-regression model.

To address possible autocorrelation issues, we considered different spatial correlation structures (exponential, Gaussian, Rational Quadratic, and Spherical) for the weighted mean effect size and the random effects meta-regression[83–85]. Based on the Akaike Information Criterion for small samples (AICc[85,86]), the exponential correlation structure was chosen (Table S7). The significance of the meta-regression model as a whole was tested by an $F$-test, whereas the significance of each partial regression coefficient (moderator) was tested using $t$-tests. All analyses were performed in R environment[78] using the metafor[84] and vegan[87] packages. All R code is provided in Appendix S3.

### Inclusion & ethics
Authors declare that all research was conducted under the necessary permissions, and that the conformation of the authors' team was specifically tailored to include geographic and demographic diversity. Invitations to join the team and participate by developing local field experiments were sent to a geographically and demographically balanced set of research teams, so the final composition and geographical coverage of the authors' list are due to the geographical distribution of the specialists that responded to the original call.

### Reporting summary
Further information on research design is available in the Nature Portfolio Reporting Summary linked to this article.

## Data availability
All the data generated in this study are provided in the Supplementary Information, as Supplementary Data S1 to S7.

## Code availability
All analyses were based on published R libraries and packages, as indicated in the Methods. The R code used to develop the analyses

data are available in the supplementary materials, as appendices S1, S2, and S3.

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

## Acknowledgements

We wish to thank Caitlin Austin, Lisbet Chaves Murcia, Renata Frederico, Leinny González, Ricardo Madrigal, Gabriel R. Navas-S, Jaime Pelayo, Pilar Salgado, Diego dos Anjos Santos, Zander Augusto Spigoloni, Victor Tituaña, Maryory A. Velado-Cano, Javier Yánez-Coronel and many colleagues and students who helped or supported field and lab work in different parts of the world. We are also indebted to the farmers and land owners that let us use their properties to conduct this study. Jean-Pierre Lumaret, Francisco Sánchez-Piñero, Ana Rey, and Andrew Sugden discussed the results and helped us find their full implications. This work was conducted with the support of the following funding sources: Spanish "Consolidando Investigación" 2015 young researchers grant (to AMCS) funded by the *Asociación Española de Ecología Terrestre*; Spanish PID2021-122380NA-I00 (DUNGPOOL) grant funded by MCIN/AEI/10.13039/501100011033 "ERDF A way of making Europe"; and Spanish "Fundación de la Universidad Autónoma de Madrid" grant 820017. Additional support came from Colombian COLCIENCIAS-PDBC No. 568 PhD scholarship (to JAN); EU H2020 research and innovation programme grant No 854248 (to FAM); and Spanish *Ramón y Cajal* fellowship RYC2020-029407-I, funded by MICIN/AEI/10.13039/501100011033, and by "ESF Investing in your future" (to AMCS). The Hungarian study was supported by the Hungarian Széchenyi 2020 Programme (VP3-16.1.1-4.1.5 No. 3043361958).

## Author contributions

Conceptualization: J.A.N., J.H., A.M.C.S. Methodology: J.A.N., J.H., A.M.C.S. Experiment coordination: J.A.N. Investigation: All authors. Funding acquisition: A.M.C.S. Analyses: I.d.C.-A., F.A.-M., J.C.G.O., L.M.B., A.M.C.S., with J.A.N., J.H. Writing—original draft: J.H., J.A.N., A.M.C.S., with I.d.C.-A., F.A.-M., J.C.G.O., L.M.B. Writing—review & editing: J.H., with J.A.N., I.d.C.-A., F.A.-M., J.C.G.O., L.M.B., N.R.A., L.A., S.B., A.L.V.D., F.E., M.E.F., K.D.F., F.G.H., R.M., T.M., B.N., C.P., A.R., C.H.S., Y.S., T.W., A.M.C.S.

## Competing interests

The authors declare no competing interests.

## Additional information

Jorge Ari Noriega [1,2,51], Joaquín Hortal [1,3,4,51]✉, Indradatta deCastro-Arrazola [1,5], Fernanda Alves-Martins [1,6,7], Jean C. G. Ortega [3,8], Luis Mauricio Bini [3], Nigel R. Andrew [9,10], Lucrecia Arellano [11], Sarah Beynon [12], Adrian L. V. Davis [13], Mario E. Favila [11], Kevin D. Floate [14], Finbarr G. Horgan [15,16], Rosa Menéndez [17], Tanja Milotic [18], Beatrice Nervo [19], Claudia Palestrini [19], Antonio Rolando [19], Clarke H. Scholtz [13], Yakup Senyüz [20], Thomas Wassmer [21], Réka Ádám [22], Cristina de O. Araújo [3], José Luis Barragan-Ramírez [23], Gergely Boros [24], Edgar Camero-Rubio [25], Melvin Cruz [26], Eva Cuesta [1,27], Miryam Pieri Damborsky [28], Christian M. Deschodt [13], Priyadarsanan Dharma Rajan [29], Bram D'hondt [18], Alfonso Díaz Rojas [11], Kemal Dindar [20], Federico Escobar [11], Verónica R. Espinoza [1,30], José Rafael Ferrer-Paris [31,32,33], Pablo Enrique Gutiérrez Rojas [34], Zac Hemmings [9], Benjamín Hernández [35], Sarah J. Hill [9], Maurice Hoffmann [18,36], Pierre Jay-Robert [37], Kyle Lewis [12,38], Megan Lewis [39,40], Cecilia Lozano [31,41], Diego Marín-Armijos [42], Patrícia Menegaz de Farias [43], Betselene Murcia-Ordoñez [34], Seena Narayanan Karimbumkara [29], José Luis Navarrete-Heredia [23], Candelaria Ortega-Echeverría [44], José D. Pablo-Cea [45], William Perrin [37], Marcelo Bruno Pessoa [1,3], Anu Radhakrishnan [29], Iraj Rahimi [46], Amalia Teresa Raimundo [28], Diana Catalina Ramos [25], Ramón E. Rebolledo [47], Angela Roggero [19], Ada Sánchez-Mercado [31,32,48], László Somay [22], Jutta Stadler [49], Pejman Tahmasebi [46], José Darwin Triana Céspedes [34] & Ana M. C. Santos [27,50]✉

[1]Department of Biogeography and Global Change, Museo Nacional de Ciencias Naturales (MNCN-CSIC), Madrid, Spain. [2]Grupo de Agua, Salud y Ambiente, Facultad de Ingeniería, Universidad El Bosque, Bogotá, Colombia. [3]Departamento de Ecologia, Instituto de Ciências Biológicas, Universidade Federal de Goiás, Goiânia, GO, Brazil. [4]cE3c – Centre for Ecology, Evolution and Environmental Changes, Faculdade de Ciências da Universidade de Lisboa, Lisboa, Portugal. [5]Departamento de Ecología, Facultad de Ciencias, Universidad de Granada, Granada, Spain. [6]CIBIO-InBIO, Research Centre in Biodiversity and Genetic Resources, University of Porto, Vairão, Portugal. [7]BIOPOLIS Program in Genomics, Biodiversity and Land Planning, CIBIO, Vairão, Portugal. [8]Programa de Pós-Graduação em Ecologia, Universidade Federal do Pará, Belém, PA, Brazil. [9]Insect Ecology Laboratory, Natural History Museum, University of New England, Armidale, NSW, Australia. [10]Faculty of Science and Engineering, Southern Cross University, Lismore, NSW, Australia. [11]Red de Ecoetología, Instituto de Ecología A.C., Xalapa, Veracruz, Mexico. [12]Dr Beynon's Bug Farm; St Davids, Pembrokeshire, United Kingdom. [13]Invertebrate Systematics and Conservation Group, Department of Zoology & Entomology, University of Pretoria, Hatfield, South Africa. [14]Lethbridge Research and Development Centre, Agriculture and Agri-Food Canada, Lethbridge, Alberta, Canada. [15]EcoLaVerna Integral Restoration Ecology; Bridestown, County Cork, Ireland. [16]Escuela de Agronomía, Facultad de Ciencias Agrarias y Forestales, Universidad Católica del Maule, Curicó, Chile. [17]Lancaster Environment Centre, Lancaster University, Lancaster, United Kingdom. [18]Research Institute for Nature and Forest (INBO), Brussels, Belgium. [19]Department of Life Sciences and Systems Biology, University of Turin, Turin, Italy. [20]Kütahya Dumlupinar University, Faculty of Art and Science, Department of Biology, Kütahya, Turkey. [21]Department of Biology, Siena Heights University, Adrian, MI, USA. [22]Centre for Ecological Research, Institute of Ecology and Botany, Vácrátót, Hungary. [23]Centro de Estudios en Zoología – CUCBA, Universidad de Guadalajara, Zapopan, Jalisco, Mexico. [24]Hungarian University of Agriculture and Life Sciences, Institute for Wildlife Management and Nature Conservation, Department of Zoology and Ecology, Budapest, Hungary. [25]Departamento de Biología, Universidad Nacional de Colombia, Bogotá, Colombia. [26]Independent researcher, Chalatenango, El Salvador. [27]Terrestrial Ecology Group (TEG-UAM), Departamento de Ecología, Universidad Autónoma de Madrid, Madrid, Spain. [28]Biología de los Artrópodos, Facultad de Ciencias Exactas y Naturales y Agrimensura (UNNE-FaCENA), Universidad Nacional del Nordeste, Corrientes, Argentina. [29]Insect Biosystematics and Conservation Laboratory, Ashoka Trust for Research in Ecology and the Environment (ATREE), Bangalore, India. [30]Facultad de Medicina Veterinaria y Zootecnia, Universidad Central del Ecuador, Quito, Ecuador. [31]Centro de Estudios Botánicos y Agroforestales, Instituto Venezolano de Investigaciones Científicas, Maracaibo, Venezuela. [32]School of Biological, Earth and Environmental Sciences, University of New South Wales, Kensington, Australia. [33]UNSW Data Science Hub, University of New South Wales, Kensington, Australia. [34]Grupo de investigación Biodiversidad y desarrollo Amazónico - BYDA, Centro de investigación Cesar Augusto Estrada González – MACAGUAL, Programa de Biología, Facultad Ciencias Básicas- Universidad de la Amazonia, Florencia, Caquetá, Colombia. [35]Departamento de Ciencias Básicas, Instituto Tecnológico de Tlajomulco, Tecnológico Nacional de México; Tlajomulco de Zúñiga, Jalisco, Mexico. [36]Terrestrial Ecology Unit (TEREC), Ghent University, Ghent, Belgium. [37]CEFE, University Montpellier, CNRS, EPHE, IRD, Université Paul Valéry Montpellier 3, Montpellier, France. [38]Pembrokeshire College, Haverfordwest, United Kingdom. [39]Harper Adams University, Newport, United Kingdom. [40]School of Biological Sciences, University of Western Australia, Crawley, Australia. [41]Instituto de Biociências, Programa de Pós Graduação em Ecologia e Conservação da Biodiversidade, Universidade Federal de Mato Grosso, Cuiabá, MT, Brazil. [42]Colección de Invertebrados Sur del Ecuador, Museo de Zoología CISEC-MUTPL, Departamento de Ciencias Biológicas y Agropecuarias, Universidad Técnica Particular de Loja, Loja, Ecuador. [43]Laboratório de Entomologia, Departamento de Ciências Agrárias e Ambientais, Universidade do Sul de Santa Catarina, Tubarão, Santa Catarina, Brazil. [44]Programa de Biología, Universidad de Cartagena, Cartagena de Indias, Colombia. [45]Escuela de Biología, Facultad de Ciencias Naturales y Matemática, Universidad de El Salvador, San Salvador, El Salvador. [46]Department of Rangeland and Watershed Management, Shahrekord University, Shahrekord, Iran. [47]Facultad de Ciencias Agropecuarias y Medioambiente, Universidad de La Frontera, Temuco, Chile. [48]Ciencias Ambientales, Universidad Espíritu Santo, Samborondón, Ecuador. [49]Department Community Ecology, Helmholtz Centre for Environmental Research, Halle (Saale), Germany. [50]Centro de Investigación en Biodiversidad y Cambio Global (CIBC-UAM), Universidad Autónoma de Madrid, Madrid, Spain. [51]These authors contributed equally: Jorge Ari Noriega, Joaquín Hortal. ✉e-mail: jhortal@mncn.csic.es; anamc.santos@uam.es

