## [Peer Review File · Nature Communications]

Dung removal increases under higher dung beetle functional diversity regardless of grazing intensificationREVIEWER COMMENTS

Reviewer #1 (Remarks to the Author):

Higher dung beetle functional diversity increases dung removal regardless of grazing intensification

Noriega et al. for Nature Communications

In this study, the authors conducted a large-scale field experiment where they assessed dung removal by dung beetles at low and high land use intensities in 38 pasturelands distributed around the globe. As diversity measures, they considered richness, abundance and functional diversity of dung beetles. They found that species richness was higher in low-intensity pastures while the other diversity metrics did not differ between low and high land use intensity sites. Differences in dung removal between low and high land use intensity sites were mainly explained by functional diversity as according to a meta-analysis. A path analysis showed that dung removal was mainly explained by dung beetle richness and abundance. The authors conclude that functional diversity increases dung removal independently of grazing regimes.

In general, the study is interesting, and the authors should be commended for such a huge field study. However, there are some issues with this manuscript pertaining to the definition of low and high land use intensity (major comment 1), the interpretation of results (see major comment 2) and the factors included in the meta-analysis (major comment 3). The path analysis seems to contradict the finding of the meta-analysis – in the former, there is no relationship between dung removal and functional diversity, while in the latter, there is a relationship between diversity – including richness, abundance and functional diversity – and dung removal. This is a central and confusing result, which is not discussed at all and challenges the main message of the manuscript. Furthermore, long parts of the discussion are not related to the results and therefore, it is difficult to follow reasoning. I feel the main text as such is difficult to understand without consulting of the supplement. However, it should be possible to follow the main text as a stand-alone document. Moreover, there are some awkward sentence structures (e.g., lines 211-216, lines 221-223, lines 251-253, lines 287-289, lines 343-345).

Major comments

1. Field experiments, low and high land use intensity: Lines 220-221: You give a reference in the supplement, but still the difference between low (< 2 animals/ha) and high (>4 animals/ha) land use intensity as you define it does not seem to be very large. Could you elaborate on why you chose this definition? Would it be possible to choose more categories – e.g., low, medium and high land use intensity – to get more differentiated results? I see that in your data, you report cattle densities of up to 100 cattle/ha, so perhaps you could differentiate more to better accommodate such extreme densities.

2. Piecewise SEM:

a) Lines 287 et seqq. and Figure 2: Could you also include Figure S5 here as a first part of Figure 2 to show all the paths you considered originally? Otherwise, it is very confusing why there is no link between climate and all diversity metrics, and the functional diversity metrics and dung removal, respectively.

b) In the supplementary methods (lines 718-719), you write that you added new paths to the initial path model: What were these new paths exactly? I am not aware that this is a common method in piecewise SEM? Do you mean the path between FDisBehavior and FDisMorphology? Why would behavioural diversity positively impact on morphological diversity? Why not the other way round? Or both?

c) I missed a discussion of the striking result that dung removal was neither influenced by behavioural nor by morphological diversity. I would have expected a positive relationship between functional diversity and dung removal. This seems to contradict the main finding of the meta-analysis which was that differences in dung removal were mainly due to functional diversity. Since this link does not exist in the SEM, the focus of the whole paper has to be shifted and the title has

to be changed. You could rather explore the relationships between richness, abundance, functional diversity and dung removal under different land use regimes (see major comment 3).

d) Lines 302-303: I do not agree. Intensification of cattle management decouples functional diversity from species richness, but not from dung removal as there was no relationship between dung removal and functional diversity, neither for low-intensity nor for high-intensity grazing.

3. Meta-analyses: Lines 674 et seqq.:

a) I am not sure the term meta-analyses is fitting since you did not really perform a meta-analysis of different studies.

b) I think it is problematic that you summarised the diversity indices abundance, richness and functional diversity since the SEM showed that there was no effect between functional diversity and dung removal. In Table 1, would it thus not make more sense to report the results separately for abundance, richness and functional diversity? I suspect you will also find positive – and perhaps stronger – effects for richness and abundance than for functional diversity. You could then shift the focus of the paper to a discussion on the impact and importance of these diversity metrics for dung removal under different levels of land use intensification, taking climate into consideration.

Minor comments

1. Normally in Nature Communications, the main text is separated into an introductory part and a “results and discussion” part.

2. Line 167: What do you mean by paired field experiments?

3. Line 183: service provisioning, change throughout manuscript.

4. Line 184/185: better: increases/decreases?

5. Line 185: regional differences in biotas: differences in biotas were not mentioned before, therefore I suggest either to remove this part of the sentence or to introduce differences in biotas beforehand.

6. Line 194: ecosystem functioning

7. Line 193: “which will increase overall biodiversity...”: Is this always the case? I would suggest phrasing this less strongly, such as: “which may increase...”

8. Lines 196-197: delete parenthesis, this information is relevant.

9. Lines 200-203: This is a very complicated and long sentence, please rephrase.

10. Line 206: What cascading effects did you study? A cascade in ecology usually pertains to trophic cascades, which you did not study here.

11. Line 207: delete parenthesis.

12. Line 208: provisioning of ecosystem services

13. Line 209: “cleansing of grazing areas”: this sounds weird and is not what dung beetles do: do you mean dung removal from grazing areas? This is the central ecosystem service this paper is about and should thus definitely be mentioned here.

14. Lines 213-214: shorten this sentence. For example: “However, in some cases high-intensity farming...”

15. Line 217: “...in 38 landscapes worldwide”: your study sites are concentrated in Europe and the

Americas and study sites are not equally distributed across the globe. I miss a discussion of this. Furthermore, the term "landscape" is vague – in 38 pasturelands may be more appropriate?

16. Fig. S1A: The explanation for the climate variable is missing.

17. Lines 221-223: awkward sentence structure, please rephrase. For example: "While dung removal constitutes a proxy...grazing density (or intensity?) is a proxy for..."

18. Line 222: grazing intensity? I suggest using uniform terms.

19. Line 224: delete parenthesis. "...such as" instead of "...e.g.,"

20. Line 243: "various effects": this is vague.

21. Line 283: for dung beetle functional diversity

22. Table 1: You could highlight the significant p value for diversity. Diversity here does not only include functional diversity, but also abundance and richness – this information is missing from the table caption.

23. Lines 289 et seqq.: I suggest reporting the results and then shortly explaining the used method.

24. Line 290: Structural Equation Modelling

25. Line 291: The driver climate was not introduced in the main text.

26. Line 292 et seqq.: use past tense when reporting the results. What are your hypotheses for the relationship between the predictors and dung removal?

27. Figure 2: The term functional diversity remains quite abstract and FDisBehavior and FDisMorphology are not explained in the text.

28. Lines 317-321: The path analysis does not show this as there is no link between functional diversity and dung removal.

29. Line 328: you did not follow a response-effect framework.

30. Lines 343-345: You did not show this.

31. Lines 345-346: You did not investigate the impact of the loss of species.

32. Lines 378-380: This is not what you show in your path analysis.

Reviewer #2 (Remarks to the Author):

Using a globally replicated experiment the authors test how intensification of cattle grazing influences dung beetles communities and dung decomposition. Such experiments are very valuable as they will allow to deduce generalities in the interplay between land use, species diversity and ecosystem functions that cannot be obtained from geographically more restricted studies.

The large international effort is laudable and in principle I think that the study design is solid. Yet, I also think that there are many alternative explanations/drivers for the results (or the heterogeneity in results) that the authors have missed to explore, even though the data should be readily available.

For example, as you are probably well aware, sampling dung beetles and measuring dung decomposition for a single 48 h period is very susceptible to variation in weather. While you account for long-term climate you also should account for the precise (e.g. temperature and precipitation) local weather conditions during each sampling.

Furthermore, the use and dosage of veterinary medication most certainly varied a lot among your 38 sites. As some (e.g. Ivermectin) can be very toxic to dung beetles and prevent dung decomposition, variation in veterinary medication can be one reason for the heterogeneity in the data. You seem to have the data and could account for this more directly. If not, this caveat should at least be more openly addressed in the writing (extending L 337).

I fully agree that the landscape and regional scale are important (L 322). You could (and probably should) account for this by including landscape variables such as forest cover or cover semi-natural habitat into your analysis. In more forested landscapes there could be spillover of large-bodied forest dung beetle species into the pastures, which could then disproportionately contribute to decomposition and hence countervail the effect of land use intensification. This is just one more example of alternative drivers for your results that you have not yet explored.

When looking at figure 1, there seems to be a latitudinal gradient in effect sizes. Have you tried to include absolute latitude (as opposed to or additional to climate) as a moderator in the meta-analyses? Doing so could also reduce the high degree of heterogeneity.

Considering that paths models test one hypothetical model structure, I think it would be important to also include the non-significant paths and variables in the compartments of figure 2 (as specified in Figure S5). For clarity, you could use thin and semi-transparent arrows and omit the path coefficients. It would also be good to report the respective test statistics of the path models, not only AIC (which tells nothing when not used in comparison).

For the trait analyses, you seem not to have corrected (or I have misunderstood the text) for the scale-dependence of morphological measurements with body size. Thus, all functional measures are strongly driven by body size and no unbiased measure of functional diversity, which could have a very large influence on all functional diversity results.

Further specific comments:

L 172: 'stocking rates' is unspecific. Somehow refer to cattle here.

L 172: Maybe explicitly name the ecosystem services you are referring to.

L 174: The last sentence of the abstract just reads as an assembly of buzzwords. It would be better to be more precise.

L 178ff: For a general non-specialist reader it would be good to define/conceptualize functional diversity in this paragraph.

L 188: I think that there are more possible scenarios than the three mentioned by you.

L 207: What is intensive and what is extensive grazing? Some more context would be necessary. Maybe you can move the text from L 220f up.

Figure 2: Are the coefficients standardized?

L 343: In several analyses of dung decomposition, dung beetle biomass (aka the share of large-bodied individuals) was a stronger driver of decomposition than diversity (e.g. Staab et al. 2022, *Journal of Animal Ecology* 91: 2113-2124; Slade et al. 2011, *Biological Conservation* 144: 166-174). Have you tested for this, as you seem to have the data (L 646)?

L 355: There are also examples for the opposite, i.e. lower dung removal in open lands than in forests (e.g. Frank et al. 2017, *Agriculture, Ecosystems and Environment* 243: 114-122; many studies from tropical landscapes).

L 385ff: The entire BEF writing is a bit superficial and could be more mechanistic, also in terms of the used terminology.

L 615: When were the samplings done?

L 619: What preservative fluid was used? Was it the same across all sites?

L 671: Which variables were transformed?

It would be valuable to have actual figures of all PCAs in the supplement, so that readers can see the different loadings on the PCs more clearly.

Reviewer #3 (Remarks to the Author):

The results are noteworthy and novel, and the work is unique in its scope. The topic is highly relevant, with cattle pastures occupying such vast areas across the globe, and understanding their effects on various ecosystem services is needed, to enable mitigation of detrimental effects. Clearly the results go beyond the focal system in regard to biodiversity–ecosystem function relationships and can guide assessment of other BEF relationships. The manuscript is clear and well-written.

What is lacking from the main text is a description of how functional diversity was measured. It is given in methods, but it is so crucial for the whole study that it should be shortly presented in the main text. Overall the two diversity measures should be more openly and critically discussed through the ms, please see below detailed notes on this.

Detailed notes on methods:

r. 598; if the dung pat was placed directly onto soil, how were you able to reweight it, being certain all of the remaining dung was included in the weighing?

r. 633-637; how were the species assigned to the food relocation groups? Was there always literature, concerning every species encountered, to refer to, for the group assignment? Did you weight each of the 12 groups equally as representing functional diversity; was a community consisting of one large, one medium and one small tunneler as functionally diverse as a community with one tunneler, one roller and one dweller (given each of the members would have had the same body lengths as for FDisBeh)? Yet, the latter community would generally be considered more functionally diverse as a dung beetle community?

r. 656-658; yet you do not discuss the complementarity anywhere? Please do add this, both in the results and in the discussion, e.g. which of the two ways of measuring functional diversity would be the one to choose, if one would use only one?

r. 648-658; while in the methods it becomes obvious that there are two definitions for functional diversity, it is less clear in the main text what is the functional diversity the results are based on.

r. 706-707; are here the FD-morph and FD-behav the same as FDisMor and FDisBeh, respectively, or something based on the latter ones?

POINT-BY-POINT ANSWER TO ALL EDITORIAL AND REFEREES' COMMENTS

The original comments from the Editor and the three reviewers are in black font.

All author's comments are in blue font in this document, and the changes to the manuscript are also in blue in the revised version

REVIEWER COMMENTS

Reviewer #1 (Remarks to the Author):

Higher dung beetle functional diversity increases dung removal regardless of grazing intensification

Noriega et al. for Nature Communications

In this study, the authors conducted a large-scale field experiment where they assessed dung removal by dung beetles at low and high land use intensities in 38 pasturelands distributed around the globe. As diversity measures, they considered richness, abundance and functional diversity of dung beetles. They found that species richness was higher in low-intensity pastures while the other diversity metrics did not differ between low and high land use intensity sites. Differences in dung removal between low and high land use intensity sites were mainly explained by functional diversity as according to a meta-analysis. A path analysis showed that dung removal was mainly explained by dung beetle richness and abundance. The authors conclude that functional diversity increases dung removal independently of grazing regimes.

In general, the study is interesting, and the authors should be commended for such a huge field study. However, there are some issues with this manuscript pertaining to the definition of low and high land use intensity (major comment 1), the interpretation of results (see major comment 2) and the factors included in the meta-analysis (major comment 3). The path analysis seems to contradict the finding of the meta-analysis – in the former, there is no relationship between dung removal and functional diversity, while in the latter, there is a relationship between diversity – including richness, abundance and functional diversity – and dung removal. This is a central and confusing result, which is not discussed at all and challenges the main message of the manuscript. Furthermore, long parts of the discussion are not related to the results and therefore, it is difficult to follow reasoning. I feel the main text as such is difficult to understand without consulting of the supplement. However, it should be possible to follow the main text as a stand-alone document.

Thank you for your review, and for highlighting these problems. It is true that our results are complex to understand, and that the former version of our work was not successful in showing how and why they are coherent, and which are the implications of these results. Your comments led us to rethink how we were communicating the analyses, and actually helped us to streamline the text.

We now realize that by putting the meta-analysis first in our initial submission, the readers could be confused by the apparent contradiction with the Structural Equation Modelling (SEM) results. The reason for this ordering was because the study was designed from the very beginning as a series of independent experiments, with the intention to analyse

them using meta-analysis to identify the most consistent driver or drivers of ecosystem service delivery (dung removal in this case). But the truth is that SEMs are quite powerful to characterize complex relationships between different factors but have important limitations when it comes to analysing heterogeneous responses. While meta-analyses are particularly powerful to assess whether there are any consistent effects along a number of otherwise heterogeneous studies (conducted in different populations, systems, etc.).

We have now shifted the order of the results such that the analyses follow a logical flow, where (i) the important predictors and the relationships between them are first identified by SEMs. However, (ii) we also identify a large degree of heterogeneity in the responses of diversity and removal to intensification in different regions, which prevents from identifying consistent drivers of function with this kind of analysis. Therefore, (iii) we use meta-analyses to assess the effect size of management regime on the different factors (climate, diversity and removal), finding a consistent effect of increased removal rates with increased morphological diversity.

We believe that shifting the order of the results to have a more logical flow will ease the understanding by the readers, thereby helping to highlight its potential importance for improving our knowledge on the functional effects of farming intensification, providing also valuable information for Biodiversity-Ecosystem Function theory. We are thus quite thankful for this comments.

Moreover, there are some awkward sentence structures (e.g., lines 211-216, lines 221-223, lines 251-253, lines 287-289, lines 343-345).

We rewrote them seeking for simplicity, often separating different statements in independent sentences (instead of the large and, arguably, complex sentences we had before). We changed these five sentences (note that line numbering has changed from former version), sometimes separating them with longer pieces of clarifying text, and revised the whole text in order to simplify it.

Major comments

1. Field experiments, low and high land use intensity: Lines 220-221: You give a reference in the supplement, but still the difference between low (< 2 animals/ha) and high (>4 animals/ha) land use intensity as you define it does not seem to be very large. Could you elaborate on why you chose this definition?

The decision of the threshold to distinguish between low- and high-intensity cattle farming was taken a priori during the design of the experimental protocol (see Noriega, Hortal & Santos 2015, ref. 63). This decision was based on our previous knowledge of cattle ranching practices, and a literature review, after which we decided to follow the criteria that were more commonly applied. An important issue to consider here is that in the literature we consulted (some of it cited in the text) there is a wide consensus that less than two animals per hectare identifies extensive management, where cattle are let to live free grazing on pasture and interventions are minimal except for occasional antiparasitic treatments, making water available through troughs or small dams, and/or provisioning of hay, green branches or other alternative food sources in periods of scarcity. In contrast, defining the minimum threshold for high-intensity ranching was not so easy, as there is a large variability between regions in the

applied practices through which intensive production is done; so although this intensification which is associated with the application of frequent deworming or parasite elimination and unnecessary antibiotic treatments, the provision of increased amounts of food to boost meat or milk production is achieved through diverse ways, including different ways to ensure the continuous supply of additional food in pastures of reduced size, combining frequent periods of stabling (yearly or weekly), and/or rapid rotation of high densities of animals in small patches of pasture, with the consequent degradation and compaction of the soil in both cases. Also, as reviewed by McAllister et al. in 2020 (cited in the paper), stocking densities vary according to climate, and may vary according to yearly variations, and ranching exploitations often combine low- and high-intensity areas in their production. Putting a threshold for high-intensity along this wide variety was quite difficult, so we ended up relying on the most common value that, according to our experience in subtropical and temperate regions, is used to distinguish low- and high-intensity by farmers in landscapes combining both practices— noting that our paired field experiments needed to be conducted in such type of landscape (albeit in a wide variety of climates). We have added an explanation of these thresholds in the first paragraph of the methods.

Would it be possible to choose more categories – e.g., low, medium and high land use intensity – to get more differentiated results? I see that in your data, you report cattle densities of up to 100 cattle/ha, so perhaps you could differentiate more to better accommodate such extreme densities.

A direct consequence of the design of our field experiments is that we can't divide our data into three levels of intensification, because they are paired: two sites per landscape. This prevents from following this advice in full. However, we have taken it into account by explicitly incorporating cow density in our SEM analyses, which have been completely reformulated. SEM results show only a non-significant relationship between cow density and dung beetle abundance (see fig. 1 of the paper). This lack of significance may in part be due to the bias towards low cow densities (as half of the sites are below 2 cows/hectare due to the design of the field experiments), but we believe that it is also because of the high heterogeneity in the responses of dung beetle communities to this factor – which in turn may be due to regional variations in both species pool and management practices. It would be indeed interesting to assess these effects at the landscape level (and some of the papers we cite go towards such direction), but we are afraid we can't go beyond with regard to these local factors with the data structure we have in this work.

In any case, we believe that the meta-analyses provide clear results about the main focus of our work, which was to assess the drivers of the heterogeneity in the responses of ecosystem service delivery to intensification. Here, higher functional diversity appears as the main factor promoting higher ecosystem service delivery in low-intensity pastures.

2. Piecewise SEM:

a) Lines 287 et seqq. and Figure 2: Could you also include Figure S5 here as a first part of Figure 2 to show all the paths you considered originally? Otherwise, it is very confusing why there is no link between climate and all diversity metrics, and the functional diversity metrics and dung removal, respectively.

We have now included the non-significant paths in the Figure (note that after text

reorganization now it is figure 1), so it is easy to see the original design of the structural model. We believe that, as the non-significant paths are now included in Fig.1, having the SEM designs also in the main text will be redundant. Thus, these designs are all placed in the supplementary materials (Extended Data figure S4, and figure S5 in the case of the alternative SEM model including latitude, see below). Of course, as this is a matter of how best to communicate our work, we would be happy to move figure S4 to the main text if the editor considers that to be more appropriate.

b) In the supplementary methods (lines 718-719), you write that you added new paths to the initial path model: What were these new paths exactly? I am not aware that this is a common method in piecewise SEM? Do you mean the path between FDisBehavior and FDisMorphology? Why would behavioural diversity positively impact on morphological diversity? Why not the other way round? Or both?

We took into account your criticisms to develop a new piecewiseSEM model. We first developed a preliminary conceptual model illustrating how we expected climate and management practices (i.e., management regimes and anthelmintic use, see Methods and supplementary data) to influence diversity metrics and dung removal. Then, we applied a d-separation test to evaluate the adequacy of the conceptual model against observational data (Shiple, Ecology 2009, <https://doi.org/10.1890/08-1034.1>). As the hypothesized conceptual model did not match the structure of relationships in the observed data, we added new paths until it fit the observed data (i.e., it reached by a d-separation p value > 0.05 , Shiple, 2009), but only if these paths were biologically plausible. This method is relatively common, and it is thought to be robust (see Bouchard et al., Ecology 2018, <https://doi.org/10.1002/ecy.2417>, now cited in the text). For example, in our initial conceptual model, we did not have a link between dung removal and taxonomic richness as we believed that this impact was indirectly mediated through behavioral and morphological diversity. However, the d-separation analysis indicated that this path was significant and we included it. Similarly, we added a path linking climate and behavioral functional diversity for the same reason. In the case of four links – between behavioural diversity and morphological diversity (1); between abundance and richness (2); and between abundance and both metrics of functional diversity (3 and 4), we added them to the model as correlated errors using the latest version of the piecewiseSEM package (which was implemented in the package piecewiseSEM 2.3.0; we updated this citation in the text). We assumed that they were produced by the same process and therefore lacked a precise causal relationship (for this implementation see Lefcheck, Met Ecol Evol 2016, <https://doi.org/10.1111/2041-210X.12512>).

c) I missed a discussion of the striking result that dung removal was neither influenced by behavioural nor by morphological diversity. I would have expected a positive relationship between functional diversity and dung removal. This seems to contradict the main finding of the meta-analysis which was that differences in dung removal were mainly due to functional diversity. Since this link does not exist in the SEM, the focus of the whole paper has to be shifted and the title has to be changed. You could rather explore the relationships between richness, abundance, functional diversity and dung removal under different land use regimes (see major comment 3).

You are right; this was confusing. As we explained above, in the former version of the paper

we were following the logical thread defined by how we designed the work from the beginning, but that was not working well. So we did two things: (i) reorganize the text putting the SEM analyses in front (as explained before); and (ii) redesigned our SEM analyses to accommodate your criticism (see next paragraph). We did not change the title because the new meta-analyses (which have been also redesigned and re-run) continue to show the same main result (i.e. that higher functional diversity increases dung removal regardless of intensification).

With regard to the new SEM design, we performed a new Piecewise SEM model including management regime as a binary variable (low-intensity = 0 and high-intensity = 1) to investigate the expected change in the response variables (*i.e.*, abundance, richness, FD-morph, FD-behav and dung removal) as land use intensifies. We also included other predictors related with intensification, including cow density (as explained above) and anthelmintic treatments (see below). We of course updated the text in the methods to explain how these SEM analyses were implemented. Importantly, the new Piecewise SEM analysis also supported the meta-regression finding that there were no significant differences in dung removal, abundance, and functional diversity between study sites with low or high intensity (see Table S6).

d) Lines 302-303: I do not agree. Intensification of cattle management decouples functional diversity from species richness, but not from dung removal as there was no relationship between dung removal and functional diversity, neither for low-intensity nor for high-intensity grazing.

The referee is right that we were overinterpreting SEM results, perhaps attributing results that in fact came from the meta-analysis. In any case, our new SEM results prove that this oversimplification was misleading. This part of the text has been completely rewritten, and of course this interpretation has been deleted, and now we make reference to a more complex situation; see lines 282-289.

3. Meta-analyses: Lines 674 et seqq.:

a) I am not sure the term meta-analyses is fitting since you did not really perform a meta-analysis of different studies.

We are confident that we can use the term meta-analysis here, following the criteria compiled by Vetter et al. (Ecosphere 2014, <https://doi.org/10.1890/ES13-00062.1>). Other important works aiming to guide the correct use of the term meta-analysis in Ecology include, for example, Koricheva & Gurevitch (J Ecol 2014, <https://doi.org/10.1111/1365-2745.12224>) and Gurevitch et al. (Nature 2018, <https://doi.org/10.1038/nature25753>).

Unfortunately, many studies use the term meta-analysis incorrectly. Specifically, meta-analyses assess the variation in effect sizes across different studies (or sites), thus fitting the following criteria, outlined by Vetter et al. (2014). First, a meta-analysis should estimate an effect size for each site (or study). Second, it is necessary to weight the effect size by its precision. Third, a weighted mean effect size should be estimated (*i.e.*, a summary effect size across studies). Fourth, confidence intervals should be calculated for each effect size as well as for the weighted mean effect size. Fifth, the heterogeneity of effect sizes between studies should be calculated. Sixth, the causes of the heterogeneity in effect sizes should be formally investigated (*e.g.*, using a meta-regression, as we do here). Seventh, the results should be

shown in a forest plot. We emphasize that all criteria were met in our work, which, as commented above, was specifically designed to be analyzed using meta-analytical techniques, as a series of independent studies using the same methodology. Note that this is equivalent to the common example of a meta-analysis based on multiple clinical studies conducted in different populations by independent teams of Medicine Doctors or researchers using comparable methodologies.

Maybe the reviewer is concerned because we did not rely on a literature search. If that is the case, we think that the following passage from Gurevitch et al. (2018) would suffice to show that our study can be correctly classified as a meta-analysis: “Meta-analysis has also been a valuable tool for practitioners in EEC [ecology, evolutionary biology and conservation] involved in collaborative research who wish to combine original results from experiments carried out across multiple study sites”. Actually, the approach of combining results from original experiments across multiple sites (which were all designed from the beginning to test a specific hypothesis) is a major strength of our meta-analysis because multiple confounding effects (e.g., different sampling methods, taxonomic resolutions, scales, etc.) were minimized or ruled out. In this sense, we emphasize that our approach can be considered a major step forward in the use of meta-analysis in ecological studies. For example, according to Ioannidis (Brit J Sports Med 2017, <http://doi.org/10.1136/bjsports-2017-097621>, cited in the text), we can classify our study as a prospective meta-analysis, which “consists in designing multiple trials with the explicit, predefined purpose to, when completed, combine them in a meta-analysis”. As mentioned above, in addition to controlling for variation due to the use of different methods (as usual and unavoidable in “traditional meta-analysis”), a major strength of a prospective meta-analysis is to be immune from selective reporting and publication bias.

b) I think it is problematic that you summarised the diversity indices abundance, richness and functional diversity since the SEM showed that there was no effect between functional diversity and dung removal. In Table 1, would it thus not make more sense to report the results separately for abundance, richness and functional diversity? I suspect you will also find positive – and perhaps stronger – effects for richness and abundance than for functional diversity. You could then shift the focus of the paper to a discussion on the impact and importance of these diversity metrics for dung removal under different levels of land use intensification, taking climate into consideration.

It is important to emphasize that the meta-analysis and SEM were used for different purposes. The former was used to model the effect sizes (as differences in dung removal between management regimes); whereas the second was used to model raw dung removal rate (before separately for each management regime, and now using a mixed model in the SEM to include management regime as a fixed factor). This was unfortunately not clear in the former version of our manuscript (our fault), but we believe that with the new organization of the text and the extensive rewriting the reasons why we have two different analyses, and what shows each one of them, are more clear.

Importantly, considering the limited number of experimental sites, it would be highly problematic to include all variables as moderators in our meta-regression model. The SEM analyses provide support in this sense, as they allow to include these other moderators. Although these analyses do not allow to compare the effect sizes that are attributable of intensification level in our meta-regression, SEM results did allow us to discard moderators

while constructing the meta-regression (which could not include many variables due to the limited number of observations). For example, we do not include anthelmintic use as a moderator because it was non-significant in the SEM (and correlated with management regime and cow density, which are considered in the meta-regression model).

In addition, we must bear in mind that some diversity indices were significantly correlated to each other (e.g., Functional Dispersion based on behavioral and morphological traits, $r = 0.64$), which would bring collinearity problems in case of using them separately as moderators in the meta-regression. Finally, as the first principal component explained a substantial part of the variance and was correlated with different variables (please, see Table S6), we would say that the use of this synthetic variable in the meta-regression model is justified, and indeed defensible.

All that said, following the reviewer’s idea (i.e., “I suspect you will also find positive – and perhaps stronger – effects for richness and abundance than for functional diversity”), we ran different meta-regression models considering as moderators the different variables in separated models (see Table R1 below). The results indicate that our previous (a priori) choice was the best one in terms of the omnibus F -test (which tests the null hypothesis that all regression coefficients are equal to zero). These exploratory results also suggest that, among the different moderators, the difference in functional dispersion based on morphological traits was the best one in predicting the effect size. Importantly, despite this more nuanced interpretation, our general conclusion that the differences in removal rates between low-intensive and high-intensity systems were mainly mediated by functional diversity was unchanged. So in short, we decided to keep using the first axis of the PCA that summarize the moderators in our meta-regression model. However, we included a new table in the supplementary material (Table S3) to show the stability of our results (considering the reviewer’s suggestion) and to highlight the (possible) relatively higher importance of functional diversity based on morphological traits. As a final remark, the stability of the results can also be verified by considering the tests we made to address one of the comments made by the second reviewer (please, see below).

Table R1. Results of a meta-regression model assessing the effects of climate ($PC1_{Climate}$) and difference in cattle density (Δ Cattle Density) on the difference in dung removal rates between management regimes (low-high intensities pastures). The effects of the other variables (differences in dung beetle abundance, richness, and functional diversity based on both behavioral and morphological traits) were tested in separate models. VIF stands for variance inflation factor. This table was included in the Supplementary Material of the manuscript (See Table S3).

Diversity metric	Parameter	Estimate	SE	t	df	P	Pseudo- R^2	F	df	P	VIF
Abundance	Intercept	0.013	0.256	0.051	34	0.960	0.090	1.001	3; 34	0.404	1.06
	$PCA1_{Climate}$	-0.156	0.102	-1.528	34	0.136					
	Δ Cattle Density	-0.203	0.250	-0.812	34	0.423					
	Abundance	-0.111	0.255	-0.436	34	0.666					
Richness	Intercept	0.054	0.251	0.213	34	0.833	0.173	2.218	3; 34	0.104	1.06
	$PCA1_{Climate}$	-0.105	0.100	-1.058	34	0.297					
	Δ Cattle Density	-0.071	0.253	-0.280	34	0.781					
	Richness	0.455	0.251	1.817	34	0.078					
FDis Behavior	Intercept	0.061	0.259	0.237	34	0.814	0.117	2.222	3; 34	0.103	

	PCA1 _{Climate}	-0.119	0.101	-1.180	34	0.246					1.03
	Δ Cattle Density	-0.209	0.244	-0.859	34	0.396					1.00
	FDis Behavior	0.408	0.220	1.857	34	0.072					1.03
	Intercept	0.132	0.268	0.494	34	0.624					
FDis Morphology	PCA1 _{Climate}	-0.116	0.104	-1.115	34	0.273	0.103	2.994	3; 34	0.044	1.03
	Δ Cattle Density	-0.282	0.238	-1.183	34	0.245					1.00
	FDis Morphology	0.520	0.227	2.293	34	0.028					1.03

Minor comments

1. Normally in Nature Communications, the main text is separated into an introductory part and a “results and discussion” part.

The manuscript was originally sent to Nature, and redirected to Nat Comm, we were following a different structure. According to the online instructions (<https://www.nature.com/ncomms/submit/article>), "the main text of an Article should begin with a section headed Introduction of referenced text that expands on the background of the work (some overlap with the abstract is acceptable), followed by sections headed Results, Discussion (if appropriate) and Methods (if appropriate)." We have followed this structure in this resubmitted version.

2. Line 167: What do you mean by paired field experiments?

We have rewritten this part of the abstract for clarity. Now it reads " We conducted paired field experiments in pastures scattered worldwide to study the effects of cattle-grazing intensification on dung removal. We measured this key ecosystem service at pastures managed with low- and high-intensity regimes within each study site, assessing between-regime differences in dung beetle communities and dung removal."

3. Line 183: service provisioning, change throughout manuscript.

We believe that in this case "provision" is the right term. "Provisioning" is used to refer to a category of ecosystem services. A provisioning service is any type of benefit to people that can be extracted from nature, including food, drinking water, timber, wood fuel, natural gas, oils, plants that can be made into clothes and other materials, and medicinal benefits. In contrast dung removal is a regulating service, which refers to the basic ecosystem processes that moderate regular phenomena, such as pollination, water purification, erosion and flood control, carbon storage, climate regulation, or as in this case decomposition. Dung removal is also part of a supporting service (i.e., those that sustain ecosystem functioning), as it is a key step of nutrient cycling in many ecosystems.

So to summarize, here and in other parts of the manuscript we say "provision" to say that the service is provided by the dung beetles. As far as we know this is common usage, and if we are not wrong the way we use it here will be directly understood by most readers; here note that some of us have worked and published about ecosystem service provision before (see e.g. Noriega et al Basic Appl Ecol 2018, cited in the text).

4. Line 184/185: better: increases/decreases?

We were trying not to repeat “increase” twice in the same sentence, but it is true that it is simpler, so we have changed enhance to increase.

5. Line 185: regional differences in biotas: differences in biotas were not mentioned before, therefore I suggest either to remove this part of the sentence or to introduce differences in biotas beforehand.

Thanks for pointing this out. As commented above, we believe that this is a central part of why responses are heterogeneous between regions, so as suggested we have introduced the origin of these regional differences through several sentences right before this statement (see current lines 189-196).

6. Line 194: ecosystem functioning

Changed as suggested.

7. Line 193: “which will increase overall biodiversity...”: Is this always the case? I would suggest phrasing this less strongly, such as: “which may increase...”

You’re right; changed as suggested.

8. Lines 196-197: delete parenthesis, this information is relevant.

We agree. Changed as suggested.

9. Lines 200-203: This is a very complicated and long sentence, please rephrase.

We have rephrased this sentence to increase its clarity, also splitting it into two different sentences.

10. Line 206: What cascading effects did you study? A cascade in ecology usually pertains to trophic cascades, which you did not study here.

Thanks for pointing this. Actually, cascade effects in ecology specifically refer to the effects of species extinctions cascading to affect the populations of other species and with it ecosystem dynamics. We were loosely talking in the broader sense of catastrophic effects that cascade along different levels of a system, but the truth is that we are not assessing the effect of extinctions (although part of the effects we observe may be due to exclusion of certain species from high-intensity management). This part of the text has been rewritten, and we have avoided using the term “cascade” from here, to avoid confusion.

11. Line 207: delete parenthesis.

Done as suggested.

12. Line 208: provisioning of ecosystem services

We have not changed this to conform with what we believe is common usage of the term; see our comment above.

13. Line 209: “cleansing of grazing areas”: this sounds weird and is not what dung beetles do: do you mean dung removal from grazing areas? This is the central ecosystem service this paper is about and should thus definitely be mentioned here.

We concur this term could be equivocal; we've changed it for "decomposing and burying dung in grazing areas".

14. Lines 213-214: shorten this sentence. For example: "However, in some cases high-intensity farming..."

We have split this sentence in two to increase clarity, so it reads as suggested, but it also keeps the expectation in the sentence beforehand. We believe that explicating this expectation is needed for the sake of the argument we make.

15. Line 217: "...in 38 landscapes worldwide": your study sites are concentrated in Europe and the Americas and study sites are not equally distributed across the globe. I miss a discussion of this. Furthermore, the term "landscape" is vague – in 38 pasturelands may be more appropriate?

We have added a short sentence explaining data coverage here, and also mentioning that they are pasturelands (current lines 224-228). When we started to write the paper we were keen on using pasturelands instead of landscapes, but after discussion with some co-authors we realized that in some region pastureland is only applied to intensively managed pastures, so we decided to use the more neutral term "landscape". After thinking about it, we have decided to leave this latter term, but could shift to pastureland if both editor and referees think it would be clearer for the broad Nat Comm readership.

16. Fig. S1A: The explanation for the climate variable is missing.

Thanks for seeing this. We have added this information, also referring to Table S5 where PCA factors loadings can be found.

17. Lines 221-223: awkward sentence structure, please rephrase. For example: "While dung removal constitutes a proxy...grazing density (or intensity?) is a proxy for..."

We have largely rephrased this entire paragraph (see current lines 219-236).

18. Line 222: grazing intensity? I suggest using uniform terms.

Thanks! Changed

.

19. Line 224: delete parenthesis. "...such as" instead of "...e.g.,"

Done.

20. Line 243: "various effects": this is vague.

This part of the text has changed, but we used "several" instead to "various" to refer to the factors that cause the high data heterogeneity we talk about.

21. Line 283: for dung beetle functional diversity

This table has changed, and so has the caption. We double-checked that all variables are well explained.

22. Table 1: You could highlight the significant p value for diversity. Diversity here does not only include functional diversity, but also abundance and richness – this information is missing

from the table caption.

As indicated in the former comment, this table has changed and so does the caption, which now includes that information.

23. Lines 289 et seqq.: I suggest reporting the results and then shortly explaining the used method.

The new text structure accounts for this. Following Nat Comm style, Methods are placed at the end, but reference to them is given in the main text, so it can be followed by the reader without necessarily having to consult the methods section.

24. Line 290: Structural Equation Modelling

We have changed this wording (now in line 251).

25. Line 291: The driver climate was not introduced in the main text.

Thanks for pointing this out. We now mention that it may affect both species pools and service provision in the first and third paragraph of the introduction.

26. Line 292 et seqq.: use past tense when reporting the results. What are your hypotheses for the relationship between the predictors and dung removal?

We prefer to refer to the analyses in past tense, and to write the results in present continuous tense (as the analyses gave these results when we performed them, but are also showing these same results to the reader now). As commented before, we included expectations for the effects of intensification and diversity on removal; we did not so for climate (except that it may have an effect) because these effects may be heterogeneous and complex, and no settled knowledge was available when we conducted the experiments; we expect that this work provides some insights about it.

27. Figure 2: The term functional diversity remains quite abstract and FDisBehavior and FDisMorphology are not explained in the text.

Thanks for pointing this. We have now added an explanation in the caption, which also refers to the methods section where the calculation of these metrics is explained.

28. Lines 317-321: The path analysis does not show this as there is no link between functional diversity and dung removal.

These results have largely changed, and this part has been removed. Also, we explain why dung removal rates tend to be higher with higher levels of functional diversity when explaining the results of the meta-analysis.

29. Line 328: you did not follow a response-effect framework.

We believe that our argument was indeed following such framework, but it may have not been clear in the former version of the paper. Thus, we have explained this further, and now this part states "Following a response-effect trait framework (ref. 33), the differences in the traits of the species present in each region may trigger different responses to these human-induced historical transformations, thereby leading to heterogeneous effects of intensification on ecological functioning." (current lines 380-383) Note that we make reference to Suding et al

Global Change Biol 2008, who presented such a framework applied to environmental change; we've now changed where in the sentence we cite this paper, to avoid confusion (before we cited it at the end of the sentence because they refer to similar issues as we do, but it is true that we are referring specifically to our results).

30. Lines 343-345: You did not show this.

31. Lines 345-346: You did not investigate the impact of the loss of species.

The concatenated sentences indicated by these two comments (30 and 31) were meant to make an argument for which we believe we have enough foundations and support from our results. We have rewritten this part to make clear that is an argument, but also where does it come from, and now it reads as follows: "Thus, according to the classical view of BEF relationships (*i.e.* higher biodiversity enhances ecosystem functioning), the adverse effects of cattle herding intensification on dung beetle diversity may produce a cascading effect that leads to a reduction in functioning associated with high-intensity cattle management as found by our prospective meta-analysis. The negative effect of intensification on ecosystem service delivery can be exacerbated if there is a selective loss of species with particular sets of traits, as indicated by the importance of functional diversity metrics in this same analysis." (lines 394-400)

32. Lines 378-380: This is not what you show in your path analysis.

We were referring to the meta-analyses, not the SEM results. We have clarified this in the text (now it reads "The results of our meta-analyses show"; line 442).

Reviewer #2 (Remarks to the Author):

Using a globally replicated experiment the authors test how intensification of cattle grazing influences dung beetles communities and dung decomposition. Such experiments are very valuable as they will allow to deduce generalities in the interplay between land use, species diversity and ecosystem functions that cannot be obtained from geographically more restricted studies.

The large international effort is laudable and in principle I think that the study design is solid. Yet, I also think that there are many alternative explanations/drivers for the results (or the heterogeneity in results) that the authors have missed to explore, even though the data should be readily available.

Thanks for making this point. You are right about there being many other factors potentially affecting the outcome of our field experiments. This creates the heterogeneity that we observe in the responses to intensification. Many of these drivers are local or regional, so including all of them explicitly would require conducting up to 38 specific analyses, one per study site. As commented above, the design of our study was specifically tailored to test the effect of low- versus high-intensity management with a meta-analysis, precisely to identify general effects that embrace all studied landscapes, going beyond such local and regional heterogeneity. Besides that, we incorporated the SEM analyses to assess the effects of several of these drivers, which are common to all sites, and were recorded in a standardized way, or at least as standardized as possible (see below, and the Methods section).

Due to these reasons, and the arguments outlined below, we believe that our study provides a fair and robust approach to the study of the effects of grazing intensification on dung beetle diversity and ecosystem functioning at the global scale. Here note that we were looking for the “bigger picture” of the general effects of management intensity. Going into more detail would have resulted into a plethora of different studies with higher knowledge of specific system but a loss of generality.

For example, as you are probably well aware, sampling dung beetles and measuring dung decomposition for a single 48 h period is very susceptible to variation in weather. While you account for long-term climate you also should account for the precise (e.g. temperature and precipitation) local weather conditions during each sampling.

Of course, dung beetle activity depends on daily weather conditions. But it also depends on the conditions of several days before, plus the conditions several months before – which determine the reproductive success of the populations of dung beetle species from the former generation that lead to the individuals that are actually active during the period of our experiments. Not to mention the large differences in seasonal regimes throughout the different regions we cover with our study sites. Accounting for this phenological variation and the eventual effects of weather on dung beetle activity and dung removal would have required performing multiple replicates of our experiment throughout one or several years, or at least throughout a season. This is obviously out of the original purpose of our work.

Rather than measuring daily variation in weather conditions, our study takes this potentially confounding factor into account by conducting dung beetle surveys and dung removal experiments almost at the same time, avoiding weeks and at days without large precipitations during the season of peak dung beetle activity (often the season when warm temperatures and precipitations coincide). Teams were instructed to conduct the surveys simultaneously or temporally overlapping as much possible to the experiments “to ensure that the structure of the dung beetle community does not change” (see the protocol at ref. 63, <http://hdl.handle.net/10261/134682>). When weather forecast indicated change in conditions, the experiments were postponed.

All the above makes sure that dung removal is functionally linked to the dung beetle community sampled by the surveys; that is, that the species active during the experiment (and their abundances) coincide with those recorded by our pitfall traps. Due to this, each paired experiment conducted in a study site (or landscape) provides a fair description of the effects of the level of intensification on both dung beetle diversity and dung removal rates. Under our analytical framework, the eventual effects of differences in weather conditions that may remain despite the efforts made to standardize our paired studies will add to the heterogeneity in the data. Such heterogeneity will add up to the unexplained variability in the SEM results, and will be explicitly accounted for by our meta-analyses. But in any case it will have a spurious effect on the parameter estimates in these analyses. We therefore believe that it does not pose any significant concern for neither the robustness nor the general relevance of our results.

Furthermore, the use and dosage of veterinary medication most certainly varied a lot among your 38 sites. As some (e.g. Ivermectin) can be very toxic to dung beetles and prevent dung decomposition, variation in veterinary medication can be one reason for the heterogeneity in

the data. You seem to have the data and could account for this more directly. If not, this caveat should at least be more openly addressed in the writing (extending L 337).

You are right in that this is an important factor directly related with the effect of management intensity on dung beetle communities. In this case it was possible to obtain data about anthelmintic treatments, so following your advice we have included this variable in our newly designed piecewise SEM analyses. Although the effects of ivermectins on dung beetles have been repeatedly demonstrated, our analyses do not show significant effects of this factor, most likely due to the large heterogeneity of management practices in our globally scattered sites. We now indicate this in the second paragraph of the discussion.

I fully agree that the landscape and regional scale are important (L 322). You could (and probably should) account for this by including landscape variables such as forest cover or cover semi-natural habitat into your analysis. In more forested landscapes there could be spillover of large-bodied forest dung beetle species into the pastures, which could then disproportionately contribute to decomposition and hence counteract the effect of land use intensification. This is just one more example of alternative drivers for your results that you have not yet explored. As commented above, studies were conducted in pasturelands where intensive and extensive management practices coexist. This includes the potential spillover of individuals between the pastures managed by these two regimes, as well as from the fragments of forest, shrubland or other open natural habitats that there may be in the surroundings. This effect is common to each studied landscape, and alike weather variations, accounting for it would require performing one separate study in each studied area, and replicates of the experiments in many patches of pasture and semiopen habitats, plus spatially-explicit descriptions of the distance of each experiment to each type of habitat. This is certainly out of the bounds of this particular paper, where we seek for general effects of intensification. So to avoid it, we tried to standardize sampling sites by avoiding the close proximity to closed habitats. As indicated in the protocol (ref. 63, <http://hdl.handle.net/10261/134682>), "the distance between the [...] sampling points to the border of any kind of forest should ideally be [at least] between 500 m to 1 km." As with weather, any eventual effects of differences in landscape configuration that may remain despite the efforts made to standardize our paired studies will add to the heterogeneity in the data. And are thus not a concern for the generalism and robustness of our results.

When looking at figure 1, there seems to be a latitudinal gradient in effect sizes. Have you tried to include absolute latitude (as opposed to or additional to climate) as a moderator in the meta-analyses? Doing so could also reduce the high degree of heterogeneity.

We did not try to include absolute latitude as a moderator in our meta-regression model because we cannot formulate a functional hypothesis that links latitude with either diversity or removal. Rather, based on current knowledge any interpretation assuming an effect of latitude would be mainly linked to climate variables. You can see a detailed discussion against the use of latitude as an explanatory variable in Hawkins & Diniz-Filho (Ecography 2004, <https://doi.org/10.1111/j.0906-7590.2004.03883.x>). But in a nutshell, even if we assume the possibility that climate (as summarized by a synthetic variable obtained through Principal Component Analysis in our case) is not exerting a direct effect on dung removal rates, and is instead acting as a proxy for other unmeasured factor(s), we are of the opinion that it would

be a more direct proxy than latitude. There are of course some latitudinal variations of diversity that can be linked to factors different to climate, such as historical processes and longitudinal biogeographical barriers, but it is unlikely that they bear direct effects on the BEF system studied here; and, importantly, we already account for this kind of effects through the inclusion of biogeographic region in our analyses. This is why we did not think about using latitude as predictor in the first place, and why we prefer not to do so now.

However, you are right that there is an evident latitudinal pattern in effect sizes (note that the figure is currently fig. 3), so to take this comment into account we performed a sensitivity analysis regarding this choice (i.e., to use latitude instead climate variables). To do this, we reran our meta-regression model again using latitude instead of climate as moderator. The results were qualitatively similar to the ones with the use of climate (Table R2) as they still indicated a significant effect of difference in diversity measures (as summarized by the first principal component axis). However, the use of absolute latitude (instead of climate) as moderator indeed resulted in a model with a higher omnibus *F*-test (which test the null hypothesis that all regression coefficients are equal to zero) and a higher pseudo-*R*². As an additional remark, we emphasize that the results of the meta-regression model with both latitude and climate as moderators were, as expected, plagued by multicollinearity (VIF values of the moderators: **4.29**, 1.02, 1.14, **4.50**, for Climate-PC1, difference in cow density, difference in Diversity-PC1 and latitude, respectively). Thus, we did not consider this model further in our manuscript.

To summarize, considering the conceptual discussion summarized above (i.e., the relative epistemic strength of latitude and climate to infer causal relationships), we decided to show both results, putting climate in the main text (Table 1 and Table S3), but reporting also the results using latitude in the Supplementary Material (Table S4, which is the same as Table R2 here). In general, we consider these tests (see also the first reviewer's comments) as sensitivity analyses, which were important to show the robustness of our main results against different model specifications. So we make the choice of continuing to put climate in front because of the functional link that we can actually make between climate and dung beetle BEF.

Table R2. Results of a meta-regression model assessing the effects of latitude and difference in cattle density (Δ Cattle Density) on the difference in dung removal rates between management regimes (low-high intensities pastures). The effects of the other variables (differences in dung beetle abundance, richness, and functional diversity based on both behavioral and morphological traits) were tested using the scores of a principal component analysis to summarize them (PC1_{Diversity}) and using each variable separately. This table was included in the Supplementary Material of the manuscript (See Table S4).

Diversity metric	Parameter	Estimate	SE	t	df	P	Pseudo- R ²	F	df	P	VIF
PCA1 Diversity	Intercept	0.816	0.536	1.521	34	0.137	0.192	3.751	3; 34	0.020	1.11
	Latitude	-0.025	0.016	-1.539	34	0.133					
	Δ Cattle Density	-0.187	0.233	-0.804	34	0.427					
	PCA1 _{Diversity}	0.351	0.163	2.158	34	0.038					
Abundance	Intercept	1.126	0.517	2.178	34	0.036	0.180	2.283	3; 34	0.097	1.08
	Latitude	-0.039	0.016	-2.486	34	0.018					
	Δ Cattle Density	-0.199	0.239	-0.833	34	0.411					
	Abundance	-0.162	0.248	-0.654	34	0.518					
Richness	Intercept	0.856	0.521	1.642	34	0.110	0.220	2.771	3; 34	0.057	

	Latitude	-0.028	0.016	-1.770	34	0.086						1.11
	Δ Cattle Density	-0.089	0.249	-0.359	34	0.722						1.15
	Richness	0.356	0.254	1.402	34	0.170						1.26
	Intercept	0.911	0.524	1.737	34	0.092						
FDis Behavior	Latitude	-0.029	0.016	-1.862	34	0.071	0.182	2.862	3; 34	0.051		1.06
	Δ Cattle Density	-0.205	0.236	-0.869	34	0.391						1.00
	FDis Behavior	0.339	0.219	1.548	34	0.131						1.06
	Intercept	0.939	0.535	1.756	34	0.088						
FDis Morphology	Latitude	-0.028	0.016	-1.751	34	0.089	0.172	3.465	3; 34	0.027		1.07
	Δ Cattle Density	-0.269	0.232	-1.160	34	0.254						1.00
	FDis Morphology	0.439	0.226	1.943	34	0.060						1.07

Considering that paths models test one hypothetical model structure, I think it would be important to also include the non-significant paths and variables in the compartments of figure 2 (as specified in Figure S5). For clarity, you could use thin and semi-transparent arrows and omit the path coefficients. It would also be good to report the respective test statistics of the path models, not only AIC (which tells nothing when not used in comparison).

Thanks for pointing this out. We have now included the non-significant paths in Fig 2.

For the trait analyses, you seem not to have corrected (or I have misunderstood the text) for the scale-dependence of morphological measurements with body size. Thus, all functional measures are strongly driven by body size and no unbiased measure of functional diversity, which could have a very large influence on all functional diversity results.

As you point out, traits are often highly correlated, and body size does correlate with many other morphological traits. That is why most functional diversity metrics use a multivariate framework, where all trait values are considered together as trait distance matrices, as the axes of variation extracted from these matrices, or as classification trees also extracted from these matrices. This strategy makes sure that the variation in all traits is considered together; of course, body size has a large effect in these distance matrices, but it couldn't be otherwise, as it covaries with many other traits. Besides such covariation, body size is also quite important for the effects of dung beetles on dung removal (see deCastro-Arrazola et al. Ecology 2020 paper we cite, ref. 19). So eliminating body size-driven variations (i.e. "correcting for body size") from the morphological diversity metrics would be "throwing away the baby with the bathwater", as we would be eliminating the direct effects of body size on functioning, as well as part of the effects of other traits that covary with body size.

Note also that, following in part the results of the experiments of ref. 19, we do account for this potential covariation by separating behavioural and morphological traits in two independent metrics, noting also that body size is part of both metrics (as dung beetle functional groups also take body size into account). We have added some clarifications and explained the reasons for this choice in the "Functional diversity measurements" section of the Methods.

Further specific comments:

L 172: 'stocking rates' is unspecific. Somehow refer to cattle here.

Thanks! We added cattle before.

L 172: Maybe explicitly name the ecosystem services you are referring to.

Here we use dung removal for a proxy of many functions (and with them services) related to it, so we added “related to decomposition and nutrient cycling”, which identify the main categories these service pertain to (the former is a regulating service and the latter is a supporting service, so quoting both we refer to the wide variety of services that may be enhanced).

L 174: The last sentence of the abstract just reads as an assembly of buzzwords. It would be better to be more precise.

We have rewritten this sentence to improve its clarity, which now reads: “This implies that actions which help maintaining functional diversity within the landscape will not only improve biodiversity conservation, but also enhance matter and energy cycles, and with it maximize ecosystem service delivery.”

L 178ff: For a general non-specialist reader it would be good to define/conceptualize functional diversity in this paragraph.

Done, see current lines 195-196.

L 188: I think that there are more possible scenarios than the three mentioned by you.

There may be, and in fact now we have added in the former paragraph a longer explanation about how biogeographical variations may affect the outcome of agricultural intensification, adding heterogeneity to the responses of both biodiversity and function. We believe that, together with the clarification in the former paragraph, now we present the main scenarios that may occur after intensification, at least according to the literature we know. Currently we don't know of any other scenario besides these three, but we will be happy to add any plausible scenario that you may suggest to this set of three.

L 207: What is intensive and what is extensive grazing? Some more context would be necessary. Maybe you can move the text from L 220f up.

This paragraph has been significantly enlarged, and we have included some clarification about what we consider high- and low-intensity management regimes (see current lines 228-236).

Figure 2: Are the coefficients standardized?

Yes; we now indicate this in the caption (currently the SEM results are shown in figure 2).

L 343: In several analyses of dung decomposition, dung beetle biomass (aka the share of large-bodied individuals) was a stronger driver of decomposition than diversity (e.g. Staab et al. 2022, *Journal of Animal Ecology* 91: 2113-2124; Slade et al. 2011, *Biological Conservation* 144: 166-174). Have you tested for this, as you seem to have the data (L 646)?

Thanks for pointing this out. We did not do so in our original analyses. Our study was not designed to measure the direct effects of biomass on function. To do that, we would have needed to either sample or control the dung beetle individuals that reached each experimental dung pat. Rather than that, we measured the abundance, richness and functional diversity of the community that is present at each site.

It could be argued that we could use the biomass recorded in the traps as a proxy for the biomass in the experiments. Thus, following your advice we calculated total biomass per

site and management regime (based on trait data and abundance, as you mention), and correlated it with removal, but we find that they are not correlated at all (Pearson $r = -0.042$, n.s.). Therefore, we discarded adding it as an additional variable in our analyses. We do not believe that this is because dung beetle biomass may not have an impact on dung removal at a global scale. Rather, it may be a matter of our design not being adequate to measure its effect, as explained above.

L 355: There are also examples for the opposite, i.e. lower dung removal in open lands than in forests (e.g. Frank et al. 2017, *Agriculture, Ecosystems and Environment* 243: 114-122; many studies from tropical landscapes).

You are completely right, thanks for stressing this. Our focus with this work has been to study open pasturelands, but it is true that here we are discussing about the colonization of new pastures by forest species, so this issue is both topical and quite interesting to mention. We have enlarged this part of the discussion, citing some new trait-based research comparing forests and open areas in different biomes, as well as several key references from Europe (incl. Frank's or Staab's above) and South America. See current lines 413-416.

L 385ff: The entire BEF writing is a bit superficial and could be more mechanistic, also in terms of the used terminology.

Here we tend to disagree; we chose the BEF theoretical framework precisely because we believe that our results add interesting information to the debate in this area, as we disentangle effects of richness and functional diversity, and show how they may interrelate in a context of (global) change. Of course this is an example with a single function and a single global change stressor (plus some moderators), but we believe that the links to the mainstream BEF literature in both the introduction and the discussion are topical, and will be useful to link our findings with this important body of knowledge.

L 615: When were the samplings done?

We have added this information to the text (before it was only available in the supplementary materials). Since surveys were conducted right after the experiments, this information is included in the Dung removal experiments subsection of the methods.

L 619: What preservative fluid was used? Was it the same across all sites?

It was 1.5 L of water (depending on weather conditions) + 4 spoons of kitchen salt (NaCl) + 2 spoon of scentless soap (preservation fluid). This information is in the protocol (current ref. 63, <http://hdl.handle.net/10261/134682>), but we added a small clarification here.

L 671: Which variables were transformed?

All of them were log-transformed (as $\log+1$) to perform the Wilcoxon tests; we have indicated that in the text now.

It would be valuable to have actual figures of all PCAs in the supplement, so that readers can see the different loadings on the PCs more clearly.

We think that the actual numbers (in a Table) are more informative. Thus, we decided to keep these results in Tables. For example, if we follow this suggestion, then, we would inevitably

need to create acronyms to inform the names of the different variables (in the caption of the Figure) given their long lengths (e.g., Minimum Temperature of Coldest Month). Besides, relationships can be quickly assessed from the factor loadings in the table. Thus, at the end of the day, we think that it would be less clear to show that a given variable is (or is not) highly correlated with the axes (please, see Figure R1 below as an example of what we mean).

We therefore decided not to include these figures in the supplement, but could do so under an editorial decision, should the editor believe they could enhance understandability.

Fig. R1. Loadings of the climate variables on the first two axes of a principal component analysis. AMT: Annual Mean Temperature; MDR: Mean Diurnal Range; ISO: Isothermality; TEMS: Temperature Seasonality; MTWM: Maximum Temperature of Warmest Month; MTCM: Minimum Temperature of Coldest Month; TAR: Temperature Annual Range; MTWQ: Mean Temperature of Wettest Quarter; MTDQ: Mean Temperature of Driest Quarter; APREC: Annual Precipitation; PRECS: Precipitation Seasonality; PWQ: Precipitation of Wettest Quarter; PDQ: Precipitation of Driest Quarter. Observation: this Figure was included only in this letter to address the reviewer’s comment.

Reviewer #3 (Remarks to the Author):

The results are noteworthy and novel, and the work is unique in its scope. The topic is highly relevant, with cattle pastures occupying such vast areas across the globe, and understanding their effects on various ecosystem services is needed, to enable mitigation of detrimental effects. Clearly the results go beyond the focal system in regard to biodiversity–ecosystem function relationships and can guide assessment of other BEF relationships. The manuscript is clear and well-written.

Thanks for your comments. We also hope this can guide the assessment of other BEF relationships to go beyond the well-documented relationship between species richness and ecosystem functioning.

What is lacking from the main text is a description of how functional diversity was measured. It is given in methods, but it is so crucial for the whole study that it should be shortly presented in the main text. Overall the two diversity measures should be more openly and critically discussed through the ms, please see below detailed notes on this.

Thanks for pointing this out. As commented above, this paper was original submitted to other journal with more limited space, but the less strict length requirements of Nat Comm have allowed us to discuss several important issues in more detail. Now the rationale behind our two functional diversity metrics is presented in the introduction (current lines 241-249), and we have also improved how we discuss them throughout the text, following your advice (see also below) and those from the other reviewers.

Detailed notes on methods:

r. 598; if the dung pat was placed directly onto soil, how were you able to reweight it, being certain all of the remaining dung was included in the weighing?

This is explained in the experimental protocol that was already cited in the text. (ref. 58): “the dung should be cleared of the pieces of earth [and vegetation] that might be attached to it, placed inside a paper bag and weighed (fresh); afterward each paper bag should be placed in a separate plastic bags and be taken to the laboratory. In the laboratory, each dung should be dried (without being removed from the paper bag) at 80 °C (176 °F) for 72 hours and weighed again (dry).” We have now added a short mention to this in the main text.

r. 633-637; how were the species assigned to the food relocation groups? Was there always literature, concerning every species encountered, to refer to, for the group assignment?

We did so according to the knowledge from the local teams, which in most cases were composed by local dung beetle specialists, as well as by many regional references. We now indicate this in this paragraph of the methods. As these references were all specific to each region, and were used by the local authors just for consultation, we did not include an exhaustive list in the paper.

There were no known problems in assigning species to functional groups for any of the 38 landscapes. Here we must note that kleptoparasitic behaviour is sometimes difficult to detect, but as far as we know that is often the case only for facultative kleptoparasites, which also show endocoprid behaviour. When a species is an obligatory kleptoparasite, that is typically included in its description, so we assume that our eventual misrepresentation of this functional group has been minimal.

Did you weight each of the 12 groups equally as representing functional diversity; was a community consisting of one large, one medium and one small tunneler as functionally diverse as a community with one tunneler, one roller and one dweller (given each of the members would have had the same body lengths as for FDisBeh)? Yet, the latter community would generally be considered more functionally diverse as a dung beetle community?

Thanks for pointing this out. No, we did not do so. As commented in the text, we used the functional classification to calculate Functional Dispersion (FDis), which is based on distances along classification trees, precisely to be able to codify these differences in the phenetic tree; behaviour was the root in the tree, with a first split between klepto/endo and para/tele behaviours, which are known to be related, and a second split in each branch to reach a tree with four tips corresponding to the four main behaviours. Which then were splitted into the body size classes. We have now mentioned this in the corresponding part of the methods.

r. 656-658; yet you do not discuss the complementarity anywhere? Please do add this, both in the results and in the discussion, e.g. which of the two ways of measuring functional diversity would be the one to choose, if one would use only one?

In the former version we were passing over the differences between both functional diversity metrics because our analyses were failing to detect any strong pattern in the way they affect dung removal. This stands out in the current version of the analyses, although there are some differences in the predictors that affect each one of them according the SEM analyses. As our focus is in how they affect function, we do not stress much such differences in the importance of climate. However, we now mention the lack of difference explicitly in the results.

r. 648-658; while in the methods it becomes obvious that there are two definitions for functional diversity, it is less clear in the main text what is the functional diversity the results are based on.

As commented above, we now explain this in the introduction.

r. 706-707; are here the FD-morph and FD-behav the same as FDisMor and FDisBeh, respectively, or something based on the latter ones?

Yes, they are, thanks for noting this. We have homogenized the terminology throughout the text.

REVIEWER COMMENTS

Reviewer #1 (Remarks to the Author):

I congratulate the authors on their careful revision of the paper. I appreciate the responses and the modifications by the authors to address my previous comments, both in cases where my concerns were integrated in the paper and in cases where the authors justified not to do so. The quality of the paper has improved considerably. I just have few comments left:

1. It would be good to have access to the R code to be able to reproduce the analysis done in the manuscript
2. Line 252: piecewise SEM as such does not show interactions, I am therefore not sure "interactions" is the best wording here. "The set of relations" would perhaps be more appropriate
3. Lines 260-266: Use simple past, it is not common to present results in present tense.
4. Fig. 1: Does each circle represent one study site?
5. Fig. 2: Fig. 2 is still Fig. 2 and not Fig. 1 in the revised version of the manuscript? Now the Figure is very complex – I meant you could show the potential paths as a Fig. 2 a) instead of Figure S4 and the actual paths as a Figure 2 b). I think this would be more appropriate but of course this is for the authors to decide.
6. Caption of Fig. 2: "taken biogeographical variation in account" – this is difficult to understand. Please add "as a random effect" so it becomes clear why this variable is not included in the path analysis.
7. Discussion in general: I still feel the discussion is poorly structured. Perhaps you could first discuss the results of the SEM and then the results of the metaanalysis in the order you presented them in the results section?
8. Line 474: Could you include a reference for the *S. laticollis* release?
9. Line 639: "in our meta-regression model"
10. Response to answer to my minor comment 3: In my opinion, "provisioning" sounds more natural than "provision" - I am well aware of provisioning ecosystem services, but this was not what is expressed here, just the gerund sounds nicer.

Reviewer #2 (Remarks to the Author):

Dear Authors,

I was curious to see this manuscript in revised form as I really think that such multi-site experimental studies are very valuable. However, I have to honestly admit that I am a bit disappointed with the extent of the revisions in light of my previous comments. Mostly you provide extensive text in the responses to my comments in order to explain why you did not do a suggested change.

I still think that it would be very important to explore alternative possible explanations for your results, even if it is only in the supplement to rule out eventual biases in the inference. I am fully aware that the intention of the study design was to test for the effect of management intensity. Nevertheless, there might still be systematic variation in the data that is independent of the paired experimental design. You would by no means need "up to 38 specific analyses" to evaluate some of the more obvious alternative explanations across (rather than within) sites. For example,

variation in weather during the 48 h sampling period would be straight-forward to test in one single analysis (across sites). If there is no relationship, perfect. However, if there is, then this would warrant at least a brief discussion of how it might confound your general inference. Similarly, if the "use of absolute latitude (instead of climate) as moderator indeed resulted in a model with [...] a higher pseudo-R²", why is there no mention of this alternative scenario in the manuscript?

Don't get me wrong, I am not questioning your initial design and rationale. Yet, at present, the analyses tell only parts of the story. I am also fully aware that you cannot present every singly analysis in detail in the main text, and many of the suggested alternatives could be in the supplement.

POINT-BY-POINT ANSWER TO ALL EDITORIAL AND REFEREES' COMMENTS

The original comments from the Editor and the three reviewers are in black font.

All author's comments are in blue font in this document. Since the revisions required in the text were either relatively minor or easy to track, this time we only present a clean, final version of the manuscript.

Reviewer #1 (Remarks to the Author):

I congratulate the authors on their careful revision of the paper. I appreciate the responses and the modifications by the authors to address my previous comments, both in cases where my concerns were integrated in the paper and in cases where the authors justified not to do so. The quality of the paper has improved considerably.

Thanks a lot for these kind words. We also believe that the revision has strengthened the paper much more than in the original submission, thanks to the reviewers' constructive assessments. Thanks also for the comments on this new version.

I just have few comments left:

1. It would be good to have access to the R code to be able to reproduce the analysis done in the manuscript

We have prepared and cleaned the R code, and now is included in the supplementary materials. Appendices S1 and S2 show the code for SEMs, and Appendix S3 is a full version of the code used for meta-analyses.

2. Line 252: piecewise SEM as such does not show interactions, I am therefore not sure "interactions" is the best wording here. "The set of relations" would perhaps be more appropriate

Changed as suggested.

3. Lines 260-266: Use simple past, it is not common to present results in present tense.

Although, here, we tend to disagree about style, it is true that such wording provides a sense of generality. Thus, as suggested, we have changed wording to simple past in all instances that we were referring to results, to make it clear that these results refer to the experiments we conducted.

4. Fig. 1: Does each circle represent one study site?

Yes, they do. We have clarified it in the figure caption.

5. Fig. 2: Fig. 2 is still Fig. 2 and not Fig. 1 in the revised version of the manuscript? Now the Figure is very complex – I meant you could show the potential paths as a Fig. 2 a) instead of Figure S4 and the actual paths as a Figure 2 b). I think this would be more appropriate but of course this is for the authors to decide.

Thank you for this suggestion. We understand why you are proposing it, but we still think that it is more direct to show just results in the main text, and the prior conceptual model as a supplement. Further, we believe it fits better with *Nature Communications* style, where

methods are presented at the end of the article. So, we have not followed this suggestion. However, for comparing the two options, we have produced the suggested figure and caption, that combines figs. 2 and S4 as a 2-plate figure (see below) . We would be happy to reverse the manner of presentation under the editor's decision.

A)

B)

Fisher's C = 20.28; $p = 0.44$

Fig. 2. Structural Equation Model identifying the causal relationships between the factors affecting dung removal rates under different cattle management regimes (low-intensity and high-intensity). A) Prior conceptual model of the relationships between climate, different aspects of biodiversity and dung removal rates under different management regimes. B) Results of the Structural Equation Model showing the causal relationships between the factors affecting dung removal rates under different management regimes, taking biogeographical variation into account as a random effect. Positive and negative effects are indicated by blue and red arrows, respectively, and standardized coefficients (standardized β) are provided within the arrows; discontinuous lines indicate non-significant relationships; two-headed dashed black arrows indicate that variables have correlated errors (see Methods); r^2 values indicate the strength of fit for each fixed factor (*i.e.*, marginal r^2); significance values: * = $p < 0.05$, ** = $p < 0.01$, * = $p < 0.001$; Fisher's C statistic is a measure of how well the model fits the observed data; similarly, a model-wide P-value > 0.05 indicates that the observed data supports the hypothesized structure of the model (see Table S1). FDisMorphology and FDisBehavior stand for the**

Functional Dispersion of, respectively, morphological and behavioral dung beetle traits. See Methods for the origin of all variables.

6. Caption of Fig. 2: “taken biogeographical variation in account” – this is difficult to understand. Please add “as a random effect” so it becomes clear why this variable is not included in the path analysis.

Added as suggested.

7. Discussion in general: I still feel the discussion is poorly structured. Perhaps you could first discuss the results of the SEM and then the results of the metaanalysis in the order you presented them in the results section?

We have reworked the discussion as suggested. It was a bit tough, and it meant re-structuring the way we present the results, starting with those across sites provided by the SEM, followed by highlighting the heterogeneity in local responses, and finally showing that, despite such heterogeneity, the effects within sites are consistent. We believe that the logical thread may be easier to follow now. Thanks a lot for this suggestion, we believe that it has improved our discussion, together with the clarification of the difference between the results across and within sites (see comments to reviewer 2).

8. Line 474: Could you include a reference for the *S. laticollis* release?

There are not yet any bibliographic references, only news releases. We have thus inserted a reference into the text comprising the webpage of Rewilding Europe where this release action is explained (<https://rewildingeurope.com/news/dung-beetle-release-highlights-the-key-role-of-small-critters-in-rewilding/>). We hope this format is okay.

9. Line 639: “in our meta-regression model”

The text did read as the referee writes, so we guess that he thought that using “our” was too informal. We have thus substituted “our” by “the”.

10. Response to answer to my minor comment 3: In my opinion, “provisioning” sounds more natural than “provision” - I am well aware of provisioning ecosystem services, but this was not what is expressed here, just the gerund sounds nicer.

Thanks for this clarification. Most of us are not native English speakers, which may be because we still think that using provisioning could be confusing. Thus, to avoid confusion, we have kept provision in the place indicated by the reviewer in the former revision.

Reviewer #2 (Remarks to the Author):

Dear Authors,

I was curious to see this manuscript in revised form as I really think that such multi-site experimental studies are very valuable. However, I have to honestly admit that I am a bit disappointed with the extent of the revisions in light of my previous comments. Mostly you

provide extensive text in the responses to my comments in order to explain why you did not do a suggested change.

Thanks for your fair comment. Here we would like to highlight that peer review processes are a form of scientific discussion, where authors and reviewers debate over the work conducted (and the conclusions taken from the results). We take this debate seriously, so we considered and pondered over each and every one of your previous concerns, as we have done with the those you provide here. So, in cases where we think that your concerns were not applicable to our results, we have provided reasoned answers why our results were not affected by such concerns. In cases where your concerns were valid and could influence our conclusions, we explain why and how we changed our conclusions and rewrote the text in accordance with these concerns. To be true, your review helped, greatly, to refine our paper, which gained in quality and balance. But to be fair, we must also say that in some instances, such as the comment below, the way your concerns were written were too inexact for us to provide a clear way of accounting for them – so they were left to our interpretation. We believe we have accounted for them in all cases.

So, to conclude, although we do understand why you may be disappointed, we believe that the reasons we provided for not making some changes, and for making others just partially, continue to stand out, and do not affect the quality of our paper. This is of course a matter of opinion, but we believe that it is a fair, and well-grounded, one.

I still think that it would be very important to explore alternative possible explanations for your results, even if it is only in the supplement to rule out eventual biases in the inference. I am fully aware that the intention of the study design was to test for the effect of management intensity. Nevertheless, there might still be systematic variation in the data that is independent of the paired experimental design. You would by no means need “up to 38 specific analyses” to evaluate some of the more obvious alternative explanations across (rather than within) sites. For example, variation in weather during the 48 h sampling period would be straight-forward to test in one single analysis (across sites). If there is no relationship, perfect. However, if there is, then this would warrant at least a brief discussion of how it might confound your general inference.

If we understand the above general comments correctly, you argue that we should evaluate systematic variation across sites, in addition to the analyses of the variation within sites provided by our meta-analyses. However, we do provide such evaluation across sites. The Structural Equation Model (SEM) does it explicitly. As you do not mention any alternative type of analysis you would like to see in the paper, we believe that this issue is well covered by our current SEM analyses. Here, please note that we did conduct the analyses on latitude that you suggested in the former version of the manuscript (mentioned below) but not those on "across sites" patterns that we thought were fully covered by our extended and reformulated SEM.

The only explicit mention you make concerns weather during the experiments, which as we argued in our former letter, was not recorded during fieldwork precisely because according to our experimental protocol weather conditions were standardized as much as possible (no significant rainfall, sunny days, temperatures between 15 and 30 degrees during daily variation). That is, our data was designed to avoid unfavourable weather variation as much as possible, to ensure that differences between experiments were due to climate, management or regional factors as much as possible. So it is not adequate to represent the

eventual effects of daily weather variations. In consequence, we believe that obtaining data from nearby weather stations for all 38 sites and seeking correlations of local daily weather with any of the variables that we explore at a global level would not provide relevant information does not seem relevant the general question we want to make.

Similarly, if the “use of absolute latitude (instead of climate) as moderator indeed resulted in a model with [...] a higher pseudo-R²”, why is there no mention of this alternative scenario in the manuscript?

Thanks for pointing out this matter. The reason we are not giving importance to latitude in the text is that, as commented in our former revision, latitude does not have a functional relationship with dung beetle diversity, management or dung removal. However, climate does show such a relationship as supported by an extensive literature. Furthermore, the latitudinal diversity gradient is, in great part, driven by the latitudinal distribution of climate on Earth. So, although the two alternative models were similar in explanatory capacity, we opted to rely on the climatic model as an explanation since it was more robust and rooted in extensive study. All that said, we conducted latitudinal analyses at your request, but we barely mention them in the main text, other than a brief mention to the preeminence of climate over latitude. As it did not affect any of our conclusions and had no place in the discussion the explanation of using climate as opposed to latitude is provided in the methods. We do, however, highlight the differences between analyses across and within sites in the discussion (see comment 7 to referee 1.)

Don't get me wrong, I am not questioning your initial design and rationale. Yet, at present, the analyses tell only parts of the story. I am also fully aware that you cannot present every singly analysis in detail in the main text, and many of the suggested alternatives could be in the supplement.

Indeed, we understand that you are not criticising either the questions we ask nor the analyses we designed to provide answers. We rather appreciate your interest in assessing the potential effects of confounding factors such as the effect of daily rainfall or photoperiod on dung removal. As commented above, although the effects on removal of weather and other specific factors merits research, it cannot be answered with our data. It would require a different experimental design that that we used. More importantly, although we may concur that this question is part of the whole story of when and why dung beetles are more or less efficient in providing their services, its effect on global gradients in the effect of intensification on dung removal will be minimal compared to climate variation. With this we are not saying that such a specific question is not interesting, but rather, that it is different from that we are asking in the present study, which we believe is of more general importance.